

# Coccolithophore responses to environmental variability in the South China Sea: species composition and calcite content

X.B. Jin[1], C.L. Liu[1], A.J. Poulton[2], M.H. Dai[3] and X.H. Guo[3]

[1] State Key Laboratory of Marine Geology, Tongji University, Shanghai 200092, China

[2] Ocean Biogeochemistry and Ecosystems, National Oceanography Centre, Southampton, UK

[3] State Key Laboratory of Marine Environmental Science, Xiamen University, Xiamen 361005, China

*Correspondence to*: C.L. Liu (liucl@tongji.edu.cn)

**Abstract.** Coccolithophore contributions to the global marine carbon cycle are regulated by the calcite content of their scales (coccoliths), and the relative cellular levels of photosynthesis and calcification. All three of these factors vary between
10 coccolithophore species, and with response to the growth environment. Here, water samples were collected in the northern basin of the South China Sea (SCS) during summer 2014 in order to examine how environmental variability influenced species composition and cellular levels of calcite content. The vertical structure of the coccolithophore community was strongly regulated by mesoscale eddies. All living coccolithophores produced within the euphotic zone (1% of surface irradiance), and *Florisphaera profunda* was a substantial coccolithophore and coccolith-calcite producer in the Deep Chlorophyll-*a* Maximum
(DCM), especially in most oligotrophic anti-cyclonic eddy centers. Placolith-bearing coccolithophores, plus *F. profunda*, and other larger and numerically rare species made almost equal contributions to coccolith-based calcite in the water column. For *Emiliania huxleyi* biometry measurements, coccolith size positively correlated with nutrients, and it is suggested that coccolith length is influenced by nutrient and light related growth rates. However, larger sized coccoliths were related to low pH and calcite saturation, although it is not a simple cause and effect relationship. Genotypic or ecophenotypic variation may also be
linked to coccolith size variation.

## 1 Introduction

Coccolithophores are an important component of marine plankton communities, contributing globally to both the organic carbon pump (biological carbon pump) and the (calcium) carbonate (counter) pump. Coccolithophores may contribute up to 10% of total chlorophyll-*a*, phytoplankton carbon and 30% to 60% of calcium carbonate (calcite or particulate inorganic carbon)
in the water-column in non-bloom conditions (Poulton et al., 2006, 2007, 2010, 2014), although higher contributions (>40%) do occur in coccolithophore blooms (Poulton et al., 2013). Coccolith-based calcite can contribute up to 80% to deep-sea carbonate fluxes (Sprengel et al., 2000, 2002; Young and Ziveri, 2000). High concentrations of the cosmopolitan coccolithophore species *Emiliania huxleyi* generate large quantities of detached coccoliths (up to $3 \times 10^5$ ml$^{-1}$, Balch et al., 1991) which are detectable from space (Cokacar et al., 2004; Raitsos et al., 2006); for example, the "Great Calcite Belt" in the
Southern Hemisphere is attributed to high particle inorganic carbon from coccolithophores (Balch et al., 2011, 2014). Two relevant issues are worthy of attention: (1) coccolithophore calcification response to ocean acidification, and (2) their ecological usage in paleoceanography and paleoclimatology.

Decreasing ocean pH (termed ocean acidification), in response to increasing atmospheric and seawater $CO_2$ levels, is a major concern for marine calcifiers such as coccolithophores, as lower pH levels (and calcium carbonate saturation levels, $\Omega_C$) may
lead to calcite dissolution and/or make the process of calcite formation (calcification) more difficult (Riebesell et al., 2000; Beaufort et al., 2011). However, conflicting results concerning coccolithophore calcification have been reported from both experimental and field studies (e.g., Riebesell et al., 2000; Iglesias-Rodriguez et al., 2008; Riebesell and Tortell, 2011; Meyer and Riebesell, 2015). A recent study by Bach et al. (2015) found that laboratory findings could be reconciled when an optimum-




type response to bicarbonate ion availability and pH where considered. In the field, different communities may respond to different combinations of elevated pH and/or nutrient availability, emphasizing the importance of species composition to community responses and to the multivariate nature of the growth environment (Poulton et al., 2011, 2014). Species-specific responses to ocean acidification are evident from laboratory work (Langer et al., 2006, 2009) and in the geological record

(Gibbs et al., 2013; O'Dea et al., 2014), with regional oceanographic settings also having an important influence (Beaufort et al., 2011; Meier et al., 2014).

Coccolithophores have a long and diverse fossil record through the last 250 Ma, with specific taxa used in biostratigraphy as paleo-ecological indicators of water-column structure and paleo-productivity. For example, the abundance of the lower photic dweller *Florisphaera profunda* is used as a proxy for the depth where nutrient concentrations increase above the low levels

observed in oligotrophic low-latitude upper waters (nutricline, or nitricline in the case of nitrate) (e.g., Molfino and McIntyre, 1990), and as a measure of paleo-productivity in equatorial regions (Beaufort et al., 1997, 2001). In the South China Sea (SCS), previous studies have examined coccolith sediment assemblages in relation to variability in the nutricline, paleo-productivity and the Asian monsoon from the late Pleistocene (Liu et al., 2008; Su et al., 2013). However, an inverse relationships between the relative abundance of *F. profunda* in surface sediments and annual average net primary production is absent in the deep

basin, and the sedimentary distribution of *F. profunda* is linked more to distance from the coast and preservation conditions in the deep basin of SCS (Fernando et al., 2007). For example, the dominance of *F. profunda* (>80%) in the deep basin is linked to dissolution (Fernando et al., 2007). In summer in the SCS, *F. profunda* inhabits water depths between 50 m to 75 m (Sun et al., 2011), much shallower than its depth distribution in the West Pacific Ocean (100-200 m; Hagino et al., 2000) or South Pacific Gyre (200-300 m; Beaufort et al., 2008). In the SCS, this depth range lies within the euphotic zone, which is typically

$109 \pm 5$ m deep in summer (Chen et al., 2007a). High *F. profunda* abundance, rather than other coccolithophores species, in the upper euphotic zone could reflect high rates of primary production in the Deep Chlorophyll-*a* Maximum (DCM) (Grelaud et al., 2012). Hence, the use of *F. profunda* to reconstruct paleo-production in the SCS is still uncertain and warrants further investigation.

The SCS is the largest marginal sea in the west Pacific Ocean, covering an area of $3.5 \times 10^6$ km$^2$ (Wang et al., 2014).

Phytoplankton production and surface circulation in the northern basin of the SCS are greatly influenced by the East Asian monsoon system. In the northern part of SCS, during the summer season (June to August), the surface water is oligotrophic and well stratified, and a stable mixed layer is developed. The mean chlorophyll-*a* concentration and primary production in the euphotic zone is $0.08 \pm 0.03$ mg m$^{-3}$ and <30 mg C m$^{-2}$ d$^{-1}$, respectively (Chen, 2005; Chen et al., 2006), with the nitricline at a depth of ~60 m (Chen et al., 2006). During the winter season (December to February), surface waters are more productive

and well mixed from the strong seasonal wind stress. Mean chlorophyll-*a* concentrations and primary production are $0.65 \pm 0.17$ mg m$^{-3}$ and 550 mg C m$^{-2}$ d$^{-1}$, respectively (Chen, 2005; Chen et al., 2006), with the nitricline much shallower at around 5 m to 20 m (Chen et al., 2006).

Mesoscale eddies are typical physical oceanographic features in the SCS (Wang et al., 2003), and significantly influence the structure of the upper water column. Cyclonic eddies in the SCS cause the thermocline to shallow and thin, while anticyclonic

eddies have the opposite effect (Chen et al., 2011). Eddy activity in the SCS are related to local wind stress curl, intrusion of the Kuroshio Current and coastal baroclinic jets (Wang et al., 2003; Hu et al., 2011). Cold-water cyclonic eddies can elevate the nutricline into subsurface waters and drive enhanced phytoplankton production at levels exceeding those in the winter. For example, the average integrated primary production inside eddies in spring and in winter is 1090 and 550 mg C m$^{-2}$ d$^{-1}$, respectively (Chen, 2005; Chen et al., 2007b). Modeling studies have also shown that cyclonic eddies are significant nutrient

sources fueling the biological carbon pump in the SCS (Xiu and Chai, 2011).

In the present study, we performed an *in situ* investigation of coccolithophores (species composition, coccolith biometry) in the upper water-column of the SCS in relation to the prevailing environmental conditions (light and nutrient availability, carbonate chemistry). The aims of this research were: (1) to examine cococcolithophore biogeography in the SCS, since only





preliminary information from the northern and western SCS is available (Okada and Honjo, 1975; Chen et al., 2007a; Sun et al., 2011); (2) to link species composition and environmental conditions in the context of potential paleo-ecological relationships; and (3) to determine how coccolith morphology responds to environmental control, especially carbonate chemistry in a low-latitude marginal sea.

**2 Materials and methods**

**2.1 Field sampling**

A total of 72 water samples from 15 stations were collected during the R/V *Dongfanghong II* cruise of the National Science Foundation (2014) - 'Deep-processes in the northern basin of the SCS'. At most stations, five depths were sampled, including 25 m, 50 m, 75 m, 100 m and 150 m (Table 1). Water samples were not collected in the upper 5 m as this was extremely

nutrient depleted, with especially low chlorophyll-*a* concentrations (http://oceancolor.gsfc.nasa.gov/cms/) in summer (Fig. 1). For each water sample, 3 L was collected via a conductivity-temperature-depth (CTD) rosette sampler and filtered through 0.45 µm pore size 47 mm diameter nitrate cellulose membrane filters (Sartorius®) under gentle pressure. The filters were rinsed to remove residual saline seawater, dried on an electric heat platform (65 ℃, 10-15 mins) and then stored in Petri dishes wrapped with aluminum foil and stored frozen (-20 ℃).

**2.2 Coccolithophore and coccolith counts**

A small piece (~0.5 x 0.5 cm) of each filter was cut out and mounted on glass slides using Norland Optical Adhesive (No. 74). Coccolithophore cell counts and species identification was undertaken using cross-polarized light microscopy (Olympus BX51). In samples with abundant coccolithophore cells, individual cells (coccospheres) were counted from at least 100 field of views (FOV, diameter of each FOV was 220 µm) up to a total of 150 to 400 coccospheres. For samples with low abundance,

around 50 extra FOVs were examined. For counts of detached coccoliths, a second piece of each filter was cut out (~0.5 x 0.5 cm) and mounted on an aluminum stub with double sided conductive carbon tape and coated with gold (see Poulton et al., 2011). A Leo 1450VP Scanning Electron Microscopy (Carl Zeiss) with SmartSEM (V5.1) software was then used to automatically capture images of consecutive FOVs from a 12 x 12 FOV (each FOV was $4.054 \times 10^{-3}$ mm$^2$) grid at a magnification of x5000, providing 144 images for analysis of detached coccolith biometry. Coccolithophore species

identification by light microscopy and Scanning Electron Microscopy (SEM) followed Frada et al. (2010), Young et al. (2003) and the Nannotax3 website (http://ina.tmsoc.org/Nannotax3/). Coccosphere and coccolith abundance was calculated using the following Eq. (1):

$$C\ (ml^{-1}) = N \times S\ /\ (A \times V) \tag{1}$$

where C is coccosphere or coccolith abundance (cells/coccoliths ml$^{-1}$), N is the number of coccospheres or coccoliths counted,

S is the filtered area (45 mm diameter) on each filter, A is the area inspected (A = number of FOV x area of 1 FOV), and V is the filtered water volume (ml).

**2.3 Coccosphere and coccolith biometry and calcite estimates**

Coccolith distal shield length (DSL) and coccosphere diameter were measured from SEM images using ImageJ software (http://rsb.info.nih.gov/ij/) following Poulton et al. (2011). Individual coccolith calcite content (calcite mass) was calculated

using Eq. (2) adapted from Young and Ziveri (2000), as in Poulton et al. (2011):

$$m\ (pg\ C) = 2.7 \times k_s \times DSL^3 \tag{2}$$

where 2.7 is density of calcite (pg C µm$^{-3}$), $k_s$ is a shape constant determined for different species and DSL is the distal shield length of each coccolith (µm). For whole coccospheres, the calcite content was estimated by multiplying the calcite mass of a single coccolith (lying flat on the upper side of the coccosphere) by an estimate of the number of coccoliths in the coccosphere

(e.g. 16 to 48 coccoliths in an *E. huxleyi* coccosphere in this study).





However, three coccolith species (*Gladiolithus*, *Calciosolenia* and *Algirosphaera*) presented in the SCS do not have $k_s$ values in Young and Ziveri (2000) or in similar coccolith calcite estimates (e.g., Knappertsbusch and Brummer, 1995; Beaufort and Heussner, 1999). Hence, for the body coccolith of *Gladiolithus flabellatus*, $k_s$ values similar to *F. profunda* (0.0016) were chosen based on their similar shapes. For *Calciosolenia murrayi*, the rhomboid-shaped coccosphere is dimorphic, having both body coccoliths and narrow coccoliths around the apical opening (Young et al., 2003). Body coccolith lengths in *C. murrayi* range from 2.2 μm to 2.6 μm, with the mean length/width ratio ~3.045 in our samples, and the thickness is about 0.2 μm from Malinverno (2004). From these morphological parameters, the $k_s$ value we estimated is 0.027. For *Algirosphaera robusta*, each coccolith contains two parts: a base and a protrusion. The former is similar to a small *Syracosphaera* coccolith, with a $k_s$ value of 0.015 (Young and Ziveri, 2000) and for the latter $k_s$ value, we calculated a cylindroid-like volume which we estimated as 0.045. Combining these two estimates gave a $k_s$ value of 0.06 for *A. robusta* in this study.

## 2.4 Environmental parameters

Seawater temperature, salinity and chlorophyll fluorescence were taken from the CTD. For stations I4, I5, I6 and I7, CTD problems led to discontinuous temperature and salinity data. Mixed layer depths (MLD) were taken as the depth where the temperature difference was >0.5 ℃ with respect to surface waters (<5 m; Painter et al., 2010), while for stations I4 to I6, the MLD were only roughly determined according to vertical temperature profiles (see Fig. 2b). Euphotic zone depth is defined as the depth to which 1% of surface irradiance penetrates. Photosynthetically active radiation (PAR) through the water column is calculated following Eq. (3):

$$PAR_Z = PAR_0 \times exp(-K_d \times Z) \qquad (3)$$

where $K_d$, the vertical diffuse attenuation coefficient, is estimated by the following Eq. (4) from Wei (2005):

$$K_d = 0.027 + 0.252 \times c_p \qquad (4)$$

where $c_p$ is the beam attenuation recorded by the CTD. Identification of eddy activity was according to the temperature sections (Fig. 2) and altimeter data on sea level anomalies (SLA) and surface water flow from the AVISO website (http://www.aviso.altimetry.fr/en/home.html).

Macronutrient (nitrate+nitrite, phosphate) concentrations were determined immediately on board with colorimetric methods, using a Technicon AA3 Auto-Analyzer (Bran-Lube). The detection limits for nitrate+nitrite and phosphate are 0.1 μmol L$^{-1}$ and 0.08 μmol L$^{-1}$, respectively. Seawater carbonate parameters (total alkalinity ($A_T$) and dissolved inorganic carbon ($C_T$)) were determined following the updated Joint Global Ocean Flux Study protocols (Dickson et al., 2007). Water samples for measurements were poisoned with saturated mercuric chloride solution and stored in dark before analysis. $C_T$ was measured on board within 2 days of sampling and $A_T$ was measured within two months. $C_T$ was measured by collecting and quantifying the $CO_2$ released from the sample upon acidification with a non-dispersive infrared detector (Li-Cor® 7000). $A_T$ was measured by potentiometric Gran titration. The accuracies of the $A_T$ and $C_T$ measurements were calibrated against the certified reference materials provided by A. G. Dickson of the Scripps Institution of Oceanography. Carbonate ion concentration, carbonate calcium saturation ($\Omega_C$) and pH were calculated by CO2SYS excel macro (Pierrot et al., 2006) from nutrients, $C_T$, $A_T$, temperature and salinity.

## 2.5 Statistical analysis

Multivariate data analysis were performed to further examine the coccolithophore composition across the study sites using PRIMER-E (v. 6.0) program (Clarke and Warwick, 2001). Before analysis, the sites of zero coccolithophore biomass and those at 150 m were removed and the absolute coccolithophore abundance data were then treated by square root-transformed. Bray-Curtis Similarity matrix was constructed with these coccolithophore biomass data and was analyzed via hierarchical cluster analysis (HCA) together with non-metric Multi-Dimensional Scaling (nMDS).

Pearson's product-moment correlations and Spearman's rank correlation were used to assess the relationship between the





coccolithophore and environmental data, and principal component analysis (PCA) was performed to evaluate the main controlling factors to the environmental parameters, using PAST software (Hammer et al., 2001).

## 3 Results

### 3.1 Physicochemical settings

A conspicuous DCM was present throughout, ranging from 50 m to 75 m in depth (Fig. 3). Total nitrogen and phosphate concentrations were below the limit of quantitation in the upper 25 m, with the nutricline at a depth of ~50 m to 75 m (Fig. 3). All stations were stratified, with shallow mixed layers, ranging from 11 m to 35 m. According to the vertical temperature profiles, SLA map and surface flows (Figs. 1b and 2), two anticyclonic eddies (labelled herein as AE) and one cyclonic eddy (CE) were present in the study area; with stations X4, X3 and J1 located in AE1, F1 and D9 located in AE2, and I1, I2, I3 and

H3 located in CE1. The nutricline and DCM mirrored variability in the temperature profiles (Figs. 2 and 3), with shallowing in the upwelling CE, and deepening in the downwelling AE. Euphotic zone depths ranged from 90 m to 100 m, except at stations I1 and I2, where the euphotic zone was ~70 m depth.

### 3.2 Coccolithophore community

The average coccolithophore cell abundance was 11.82 cells ml$^{-1}$, ranging from 0 to 83.67 cells ml$^{-1}$ across the sampling sites.

The highest cell abundance was found at station I3 at a depth of 50 m. At each station, the lowest cell abundances were found at 25 m and/or 150 m, whereas the depths with the highest abundances was at 50 m and/or 75 m, in close proximity to the nutricline and DCM. A total of 17 coccolithophore taxa were counted (Table 2) across the study site.

The nMDS ordination (Fig. 4) shows that at a level of 40% (dis)similarity in the HCA, three groups occurred: Group 1 contained *Emiliania huxleyi*, *Umbellosphaera irregularis* and *Discosphaera tubifera*, with the lowest average cell

concentrations of all the groups identified (8.57 cells ml$^{-1}$), and represented the shallowest samples (25 m and 50 m). Most of the samples were located at 25 m, and some at 50 m, (Fig. 5), and were representative of oligotrophic conditions in the upper mixed layer. Group 2 contained *Emiliania huxleyi*, *Gephyrocapsa oceanica* and *Oolithotus fragillis*, with the highest average cell concentration (27.38 cells ml$^{-1}$) of all the groups. Samples in this group were usually located at depths between 45 m and 75 m (Fig. 5), around 25 m below the MLD and representing the DCM, with elevated nutrients. Group 3 included taxa

representing the lower photic zone (*Algirosphaera robusta*, *F. profunda*), with *E. huxleyi* also present in some samples. Samples in Group 3 were found at 75 m and 100 m depth (Fig. 5) in which mean cell concentrations were 17.43 cells ml$^{-1}$ and 9.04 cells ml$^{-1}$, respectively. *E. huxleyi* was the dominant species throughout the study area, occurring in nearly all samples collected (Table 2). Two distinguishable morphotypes of *E. huxleyi* (type A and type B) were observed in the SEM images, with morphotype A comprising >90% of total cell numbers.

### 3.3 Estimates of coccolith and coccosphere calcite

The mean concentration of detached coccoliths was 158 coccoliths ml$^{-1}$, with a range from 0 to 673 coccoliths ml$^{-1}$. The highest detached coccolith concentration was observed at station F1 at 75 m, corresponding to the highest cells number (22.87 cells ml$^{-1}$) at this station. However, this pattern was not common at most stations where the depth of highest cell concentration rarely corresponded to the depth with the highest detached coccolith concentration. For example, the second highest detached

coccolith concentration (623 coccoliths ml$^{-1}$) was found at station D9 at 150 m, the easternmost station sampled (Fig. 1), where coccosphere concentration was low (1.87 cells ml$^{-1}$). It is unlikely that such high abundances of detached coccoliths in deep layers of the water column could be produced *in situ* when cell abundances are so low, and hence these features may be characteristics of either lateral transport or active sinking.

Based on coccosphere and detached coccolith concentrations, estimated total calcite concentrations ranged from <0.1 to 5258.1

40  pg C ml$^{-1}$, with a cruise average of 1508.3 pg C ml$^{-1}$. Estimated total calcite concentrations roughly mirrored detached coccolith





concentrations (Fig. 6; Spearman's rank correlation, $r_s = 0.81$, p<0.01, n = 67), highlighting the contribution of detached coccoliths to particulate calcite in the water column. Our estimated calcite concentrations were in the same range as those estimated by Beaufort et al. (2008) in the southeast Pacific (2224 pg C ml$^{-1}$ on average). The relative contribution to water column calcite of three coccolithophore species (*E. huxleyi, F. profunda, Gephyrocapsa* spp.) who dominate surface sediments

(Cheng and Wang, 1997) and deep-sea calcite fluxes (Jin et al., in prep.) was also estimated. *Emiliania huxleyi* (13%), *F. profunda* (29%) and *Gephyrocapsa* spp. (8%) contributed, on average, to around 50% of water column calcite concentrations. The depth distribution of these species contributions to total calcite matched well with their average depth distribution across the study area; *E. huxleyi* contributions were highest in the upper water column (25 m and 50 m), while *F. profunda* contributions were highest at depth (75 m and 100 m).

**3.4 *Emiliania huxleyi* biometry**

A total of 2560 detached *E. huxleyi* coccoliths (morphotype A) were measured from the SEM images across the study site (n = 10 to 216 for individual samples). *Emiliania huxleyi* distal shield lengths (DSL) for each sample (station, depth) are presented as box-plots in Figure 8. From all the samples analyzed, the average DSL was 2.96 μm, with an overall standard deviation of 0.39 μm. There was no obvious environmental trend across the study area. In order to examine potential relationships with

15 environmental factors, Pearson's product-moment correlations were calculated between average DSL, nutrients (nitrite+nitrate, phosphate) and carbonate chemistry (pH, $\Omega_C$ and $A_T$) (n=29, Table 3). Statistically significant (p < 0.01) correlations occurred between DSL, total nitrogen (nitrite+nitrate) and phosphate (positive), and pH and $\Omega_C$ (negative), whereas not correlation occurred with $A_T$ (p = 0.13).

The diameter of intact coccospheres, as well as the number of coccoliths per coccosphere was measured from 102 intact *E.*

*huxleyi* coccospheres. The mean coccosphere diameter across all those measured was 6.41 μm, with a standard deviation of 0.95 μm. The average number of coccoliths estimated per coccosphere was 32, with an overall range from 16 to 48. Coccosphere diameter showed a statistically significant (Spearman's rank correlation, $r_s = 0.71$, p<0.01, n = 102) positive relationship with DSL and coccolith number (N) per sphere (Fig. 9); binary regression equation, coccosphere diameter = 1.205 × DSL + 0.106 × N + 0.096. Estimated coccosphere diameter predicted using this regression equation showed good

agreement with that measured (y = 0.955 x, $R^2$ = 0.83, p<0.01).

**4 Discussion**

**4.1 Coccolithophore biogeography in the South China Sea**

In the context of the coccolithophore biogeographical zones of Winter et al. (1994), the coccolithophore assemblages in the SCS belong to the *tropical* zone, being dominated by *E. huxleyi, G. oceanica, G. ericsonii, Oolithutos fragilis, U. irregularis,*

*F. profunda* and *A. robusta. Reticulofenestra sessilis* was also found in the SCS, and this species is exclusively found in the *tropical* zone where it may form symbioses with diatoms (e.g., *Thalassiosira* species) (Winter et al., 1994; Jordan, 2012). The coccolithophore assemblages of the SCS are similar with those in the equatorial Pacific Ocean (Hagino et al., 2000).

The two dominant species in our samples from the SCS were *E. huxleyi* and *F. profunda*, species representative of the upper and lower photic zone floral groups (Winter et al. 1994). In the SCS, these floral groups live in the euphotic zone (>1% surface

irradiance) which is ~100 m in summer (Sun et al., 2011). However, in the equatorial and subtropical gyres of the Pacific and Atlantic Ocean, these coccolithophore species are found much deeper (150 m to 250 m) in the water column (Hagino et al., 2000; Boeckel and Baumann, 2008; Beaufort et al., 2008). These differences are undoubtedly linked to differences between the SCS and open-ocean in terms of the depths of thermocline and nutricline, implying that the SCS is more eutrophic when compared with tropical settings at the same latitude.

*Upper photic zone (UPZ) assemblage*: In our nMDS analysis, the UPZ assemblage (Winter et al., 1994) was represented by Groups 1 and 2, found at 25 m and 50 m in the SCS. These two groups have different species composition in our analysis; for





example, Group 1 included umbelliform species, such as *U. irregularis*, which are considered K-selected (specialists) species and agrees well with previous work (e.g. Okada and Honjo, 1975). The UPZ assemblage is commonly observed in well stratified, oligotrophic, warm surface waters in the West Pacific Warm Pool (Hagino et al., 2000). In the SCS, *U. irregularis* was mostly found at stations with deep mixed layers, deep nutriclines and extremely low nutrients in surface waters.

In comparison, Group 2 occurred at stations with shallower mixed layers and nutriclines, and hence potentially elevated nutrient supplies, and was more diverse, with *E. huxleyi* dominant. These results contradict with other studies in the SCS in summer, such as Okada and Honjo (1975) who found that *G. oceanica* was the dominant species (30% to 100% of total cell numbers) (see also Sun et al., 2011) in the western and southern parts of the SCS. Differences between this study and others could relate to the influence of the Asian summer monsoon on the western and southern SCS, where the southwestward wind

causes a wind driven upwelling system off the east coast of Vietnam (Liu et al., 2002; Xie et al., 2003; Ning et al., 2004). *G. oceanica* is considered a more eutrophic and coastal species (Andruleit and Rogalla, 2002; Andruleit et al., 2003) and hence it contributed less to coccolithophore biomass in the central and northern part of SCS, where summer monsoon induced upwelling is weak.

Morphotype A was the dominant morphotype of *E. huxleyi* in the SCS. Four common mophotypes of *E. huxleyi* are currently

recognized in the literature (A, B, C, B/C), which can be distinguished by coccolith characteristics such as DSL, element widths and features of the central area (e.g., Young et al., 2003; Poulton et al., 2011), and may be considered as different ecotypes with different temperature and nutrient preferences (Cook et al., 2011; Poulton et al., 2011; Saavedra-Pellitero et al., 2014; Young et al., 2014). In the equatorial Pacific, *E. huxleyi* type A was abundant in fully mixed waters, while the smaller type C occurred below the thermocline (Hagino et al., 2000), and type A and type B dominated in the warm Kuroshio and cold

Oyashio currents, respectively, off Japan (Hagino et al., 2005). In the Pacific sector of the Southern Ocean, *E. huxleyi* type A was found in the subantarctic zone, while type B, C and B/C were found further south (Saavedra-Pellitero et al., 2014). In general, *E. huxleyi* type A shows a warmer water preference than type B and other type B derivatives (C, B/C).

*Lower photic zone (LPZ) assemblage:* In our study, the LPZ was represented by Group 3, which included typical LPZ species (*F. profunda*, *A. robusta* and *Gladiolithus* spp.) and was found between 75 m and 100 m. Group 3 occurred above, at or near

the depth where 1% of surface irradiance penetrated (i.e., base of the euphotic zone). In other tropical oceans, the LPZ assemblage dwells deeper than the base of the euphotic zone (Hagino et al., 2000; Boeckel and Baumann, 2008; Beaufort et al., 2008). In the northern Arabian Sea, *F. profunda* inhabits shallower waters, and is found across a wider depth range (10 m to 80 m) (Andruleit et al., 2003). It is worth noting that, as in the SCS, the Arabian Sea is strongly controlled by a monsoonal system and is considered eutrophic (Andruleit and Rogalla, 2002; Andruleit et al., 2003). Hence, we suggest that neither water

depth or light availability are limiting factors for *F. profunda* (and/or other LPZ species) in the SCS, but rather nutrient availability is important; the nitricline is shallow (50 m to 75 m) even in the oligotrophic summer in the SCS.

**4.2 The response of coccolithophores to eddies in the South China Sea**

Mesoscale eddies have a strong influence on productivity and ecosystem structure in the SCS (Chen et al., 2007b; Lin et al., 2010; Zhang et al., 2011). Previous measurements in the SCS have shown that integrated primary production in cyclonic eddies

can be 2-3 fold higher relative to the outside of eddies (Chen et al., 2007b). Modelling results have also highlighted how new production, relative to outside of eddies, can be ~30% higher or lower in cyclonic or anti-cyclonic eddies, respectively (Xiu and Chai, 2011), with nutrient upwelling along the edges of anti-cyclonic eddies and elevated particulate organic carbon export (Zhou et al., 2013).

In the context of eddies in the SCS, further examination of the nMDS ordination (Fig. 4) showed that upper ocean samples (25

40   m, green triangles) from anti-cyclonic eddies were mainly found in Group 1, while the samples from cyclonic eddies were in Group 2. The same trend was seen for deeper samples (45 m to 65 m, blue squares; 75 m, pink diamond from J1). The abundance of each of the floral groups identified (Fig. 5) clearly showed that in the anti-cyclonic eddies Group 1 and Group 3





were found in the surface and deep waters, respectively, while in cyclonic eddies the sequence was from Group 2 and Group 3. At the stations outside of the eddy, Group 1 and Group 2 assemblages both appear in the UPZ, while Group 3 occurred at all stations. In the anti-cyclonic eddy samples, LPZ coccolithophore species, in association with the DCM, were highest in abundance. This implies that *F. profunda* is not only a proxy of the nutricline, but could play an important role in deep primary

production in unproductive seasons. Hence, high abundances of *F. profunda* are not only an indicator of low productivity, but also track sub-surface production which cannot be observed by satellites (Grelaud et al., 2012; Malinverno et al., 2014). However, this paradigm is not suitable for monsoonal seasons due to the dramatic changes in environmental conditions that occur during the winter monsoon season in the northern SCS, when the water column is fully mixed and there are high nutrient levels (Tseng et al., 2005; Wong et al., 2007).

Station I5 had another distinctive arrangement of species assemblages which was opposite to that found at the other stations sampled; Group 2 was found at 25 m while Group 1 was at 50 m. Examination of the temperature profile shows that the 29.5°C isotherm was shallow and domed, while the 22.5°C isotherm was pushed deeper into the water column (Fig. 2b). Filters collected at 25 m and 50 m from I5 had lots of diatom fragments, and relatively elevated coccolithophore abundances (21.75 and 22.59 cells ml$^{-1}$ in 25 and 50 m, respectively). We suggest that this feature represents a mode-water eddy, as described by

McGillicuddy et al. (2007) in the northeast subtropical Atlantic Ocean. McGillicuddy et al. (2007) observed elevated phytoplankton production (i.e. a diatom bloom) in a mode-water eddy, which led to local changes in the zooplankton community composition (McGillicuddy et al., 2007; Eden et al., 2009). Mode-water eddies are anticyclonic, caused by interactions between local wind patterns and eddies (McGillicuddy et al., 2007), and examination of Figure 1 shows that I5 is associated with anticyclonic surface flows. In the central and southern basins of the SCS during summer, the northward

monsoon and its associated wind patterns may account for the mode-water eddy.

**4.3 Calcite concentrations in the South China Sea**

The discrete estimates of bulk coccolith calcite roughly co-varied with coccolith and coccolithophore concentration in water column, with peak concentrations around the DCM. Rather than controlled by the environmental factors (light, nutrients, carbonate chemistry), the vertical distribution of bulk coccolith calcite reflected changes in the coccolithophore community

composition. In addition, excluding the maximum calcite concentration in the DCM, another peak was also found in deeper water at some stations, for example at 150 m in F1 and D9, and 100 m and 150 m in I7 station, where the living cells were low and calcite was nearly all contributed by detached coccoliths.

Previous research in the SCS has shown that *Gephyrocapsa* dominates surface sediments (Cheng and Wang, 1997; Fernando et al., 2007), although our data (Fig. 7) shows that this taxa contributes relatively little to water column calcite concentrations.

This discrepancy in *Gephyrocapsa* contributions to upper ocean and deep calcite inventories may relate to two factors: firstly, the larger and heavier species *G. oceanica* has relative low abundances in the SCS basin; secondly, the smaller and lightly calcified *G. ericsoni* is more abundant in the SCS.

Three species (*E. huxleyi, F. profunda* and *Gephyrocapsa* spp.) represented around half of the calcite in the water column (Fig. 7), whereas other species with smaller levels of abundance contribute to the other 50% of water column calcite. The importance

of these relatively less abundant species in calcite inventories is partly related to higher per coccolith calcite contents, due in part to larger coccolith lengths and calcite contents (Young and Ziveri, 2000); for example, *Oolithus fragilis* has >80 pg C per coccolith whereas *E. huxleyi* has ~2 pg C per coccolith. Hence, relatively rare coccolithophore species with high coccolith and coccosphere calcite contents are important vectors of both upper-ocean calcite production (Daniels et al., 2014) and deep-sea calcite fluxes (Ziveri et al., 2007). However, in the case of the northern SCS basin, examination of sediment trap materials

shows that the three species dominating upper ocean calcite inventories (*E. huxleyi*, *F. profunda*, *Gephyrocapsa* spp.) also represent >95% and >80% of deep-sea coccolith and coccolith calcite fluxes (Jin et al., in prep.). This highlights the discrepancy of coccolith calcite species between euphotic zone and deep layers.




### 4.4 Environmental influences on *E. huxleyi* biometry

*Nutrients and light:* A positive relationship between nutrients (both nitrogen and phosphorus) and *E. huxleyi* calcification was found in this study (Table 3). However, some culture experiments have shown that nutrients may exert little influence on coccolith calcification or morphological variance (e.g. malformation) (Paasche, 1998; Fritz, 1999; Langer and Benner, 2009; Langer et al., 2012). And some culturing and modeling studies showed divergent responses of coccoliths calcification to nitrogen and phosphorus stress respectively (Müller et al., 2008; Aloisi, 2015). In mesocosm enclosures coccolith size has been shown to change under low phosphate conditions (Båtvik et al., 1997; Engel et al., 2005). For shipboard samples, some authors argue that coccolith malformation, and hence less calcification, can be caused by nutrient limitation (Okada and Honjo, 1975; Kleijne, 1990; Yang et al., 2004). It is interesting that these results come from different methodologies, with our results agreeing well with the field samplings apparently. However, the difference is that at most stations the largest coccoliths occurred at deepest depth, and within the *E. huxleyi* abundant layer coccoliths were relatively small (e.g. most remarkable at X3, F1, D9, I7, X5, Fig. 10a).

If nutrients are the only limiting factor in *E. huxleyi* growth (i.e. under culturing conditions), when nutrients are replete, *E. huxleyi* growth is fast (exponential phase), with fewer and smaller coccospheres per cell. When nutrients become limiting, *E. huxleyi* growth slows (stationary phase), and larger and multi-layer coccospheres are produced (Gibbs et al., 2013; O'Dea et al., 2014). A culturing study of *E. huxleyi* strain NIES 837 has shown that during rapid cell division phase, coccoliths production on cells was ceased (Satoh et al., 2009). In our cases in the SCS (in field conditions), nutrients were not the only limiting factor influencing *E. huxleyi* growth. We propose that light is also a limiting factor for *E. huxleyi* production and calcification in natural communities. Although some authors stated that light should not be regarded as a factor in phytoplankton growth in oligotrophic SCS, because the euphotic depth exceeds the MLD and nutricline throughout the year (Tseng et al., 2005; Wong et al., 2007). Here, a simple schematic is proposed (Fig. 10b): we suggest that nutrients are not limiting below the nutricline. (1) In the DCM layer, when light and nutrients are optimal for phytoplankton growth (exponential phase), *E. huxleyi* growth is fast and they produce less calcified coccoliths; (2) In deeper waters, when nutrients are more sufficient but light is not available, *E. huxleyi* growth slows and they produce heavier coccoliths (stationary phase). That light limitation, in *E. huxleyi* cell, can prolong G1 assimilation stage during which calcification take place will at last increase cellular calcite content (Müller et al., 2008); (3) Above the nutricline, when light is sufficient and nutrients are depleted, we suggest that *E. huxleyi* produces larger cells as well, though we have not enough data to support this contention as *E. huxleyi* coccoliths were too few in the SEM images in these samples for statistical significance.

The same trend of calcification in the water column has also been found off the Loffoten Islands in the Norwegian Sea (Charalampopoulou et al., 2011). Cell calcification rate was <1 pmol C cell$^{-1}$ d$^{-1}$ in the coccolithophore maximum layer, while it was about three times higher in upper and lower waters where coccolithophores were less abundant (Charalampopoulou et al., 2011), although bulk calcification in this study was influenced by light and coccolihophore species changes (Charalampopoulou et al., 2011). That opposite results were found in Benguela coastal upwelling system where coccospheres and coccoliths in the DCM (~17 m) were larger than those at 50 m depth could be due to different bloom stage of *E. huxleyi* (Henderiks et al., 2012). Largest coccoliths/coccospheres were reported in late exponential growth stage (11[th] day) in mesocosm experiments (Engel et al., 2005). It should be noted that in their experiments phosphate was exhausted at the 11[th] day (<0.05 µmol L$^{-1}$), while nitrate was not below detection limit until 13[th] day (Engel et al., 2005). It means that phosphorus limitation regulated growth rate (decrease) with co-variation of cellular calcification (increase, negative response) (Müller et al., 2008; Aloisi, 2015). However, it is not the case in the SCS, because nutrients were replete at deeper depth, and growth rate was, we suggested, limited by light availability. Results from sediment traps showed that mean coccoliths weight of *E. huxleyi* was linked to primary productivity in bloom seasons in tropical Atlantic and the Mediterranean Sea (Beaufort et al., 2007; Meier et al., 2014). Nevertheless, these changes may account for the seasonal overturn of heavily and lightly calcified *E. huxleyi* (different morphotypes) (Triantaphyllou et al., 2010; Meier et al., 2014).



Coccolithophore cell calcification is a strongly light-dependent process (Poulton et al., 2007; 2010; 2014). It seems paradoxical that light constrains coccolithophore growth rates and promotes cellular calcification, since photosynthesis and calcification are coupled (Paasche, 2001; Rost and Reibesell, 2004). It is worthy to note that, cell/coccolith size variations are a combination of physiological responses to environmental constraints, for example the different energy allocation between cell growth and

maturity under different nutrient (phosphorus and nitrogen) limitation (Aloisi, 2015), and may also be influenced by zooplankton grazing in nature conditions (Beaufort et al., 2007). Whether this response is positive or negative needs more detailed study.

*Carbonate chemistry:* Coccolithophores are thought to be sensitive indicators of carbonate chemistry, especially $\Omega_C$ and $[CO_3^{2-}]$ (e.g., Beaufort et al., 2011). High concentrations of $H^+$ will also disturb the physiology of the coccolith vesicle that precipitates

coccoliths, potentially leading to increases in coccolith malformation (Bach et al., 2012). Our results show an inverse correlation between DSL and pH, and $\Omega_C$. Similarly, *E. huxleyi* calcification has been found to be negatively correlated with $\Omega_C$ in the shelf waters of the Northwest European shelf (Poulton et al., 2014). In their case, the range in $\Omega_C$ and pH values were small compared with many open-ocean situations (Poulton et al., 2014). In our case, except $A_T$, all the environmental data was significantly correlated (Table 3), and all these parameters contribute to one principal component (Table 4), which is also depth

dependent. Importantly, in the data from the SCS the carbonate chemistry inversely mirrors the nutrient data. Hence, it is not necessary to directly infer that *E. huxleyi* calcification and carbonate chemistry have a simple cause and effect relationship in the SCS.

Here, our DSL results in the SCS were compared with those in the North Sea (Young et al., 2014) (Fig. 11). In the North Sea, *E. huxleyi* was also dominated by morphotype A (Young et al., 2014). $\Omega_C$ in the two regions falls within a similar range,

however DSL is much larger in the North Sea (*t*-test, $p < 0.001$). The morphotype in both the North Sea and SCS was A, and hence what causes the morphological distinction may be genotypic variation or an "*ecological*" effect (Bach et al., 2012). It is suggested that the changing environmental conditions can select for different coccolithophore strains, phenotypes and/or genotypic transition, which indirectly influences the coccolith size and morphology (Bach et al., 2012). For example, different environmental provinces can shift from a community dominated by normally calcified *E. huxleyi* type A to one characterized

by weakly calcified B/C in the Patagonia Shelf and Southern Ocean (Cubillos et al., 2007; Poulton et al., 2011). Heavier calcified morphotypes during low $\Omega_C$ in winter may be responsible for the seasonal morphotype transition in the Bay of Biscay (Smith et al., 2012). Seasonal variability of *E. huxleyi* coccolith size has been observed in the Aegean Sea, which may be due to ecophenotypic variation (Triantaphyllou et al., 2010). Additionally, Young et al. (2014) have argued that *E. huxleyi* DSL differences related to neritic and oceanic (phenotypic) groups rather than carbonate chemistry impacts. DSL in our samples

show no difference with those in the oceanic group (*t*-test, $p = 0.99$), however, they are significantly lower than those in the neritic group (*t*-test, $p < 0.001$) (Fig. 11). The ecological transition of assemblages may be a more dominant effect on coccolith morphology and/or cellular calcification in not only the present ocean, but also in geological records.

## 5 Conclusions

In the South China Sea (SCS), the coccolithophore community corresponds to the tropical biogeographic zone, with many

characteristic tropical species being present (e.g., *Umbilicosphaera irregularis*, *Florisphaera profunda*). Coccolithophore cellular abundances ranged from <1 cells $ml^{-1}$ to 83.67 cells $ml^{-1}$ across the SCS basin. Highest cell concentrations occurred in the DCM, with all of the coccolithophore community within the euphotic zone (i.e. above the depth where 1% of surface irradiance penetrates). *Emiliania huxleyi* (type A) was the numerically dominant species in the SCS during summer.

The coccolithophore communities showed strong responses in terms of species composition to mesoscale eddy features of the

SCS (Fig. 12). Vertical coccolithophore communities showed strong vertical diffentiation, from upper photic zone (UPZ) species to lower photic zone (LPZ) species. The species composition of the UPZ and LPZ differed between anticyclonic eddies and cyclonic eddies (Fig. 4), whereas all species occurred in non-eddy conditions.



Estimates of calcite concentrations in the upper water column based on coccosphere and coccolith calcite contents closely matched detached coccolith concentrations highlighting their significant contribution to calcite standing stocks. Three key species contributed roughly half (Fig. 7) of the upper ocean calcite concentrations (*E. huxleyi*, *Gephyrocapsa* spp., *F. profunda*), with these three species dominating deep-sea calcite fluxes (Jin et al., in prep.). The deep-dwelling species *F. profunda* showed

a shallower depth distribution (75 m to 100 m) than observed in the open-ocean of the Pacific (>150 m), and was the numerically dominant species in anti-cyclonic eddies.

Biometric measurements of *E. huxleyi* coccoliths showed significant ($p<0.01$) positive relationships with nutrient (nitrate, phosphate) concentrations and negative relationships with carbonate chemistry (pH, $\Omega_C$) (Table 3). One interpretation of these patterns is that environmental factors influence coccolith size (and hence calcification), however another is that the DSL is

different for sub-populations of *E. huxleyi* in the SCS, who occupy slightly different environmental conditions. Previous studies support both interpretations, in terms of environmental controls on coccolith DSL (e.g., Bollmann and Herrle, 2007; Horigome et al., 2014) and the biogeography of *E. huxleyi* coccolith morphologies (e.g., Poulton et al., 2011; Young et al., 2014). In the future, isolation of novel *E. huxleyi* strains (and other species) from the SCS, followed by in-depth physiological, molecular and biometric examination, will allow further insights into coccolithophore ecology in the SCS.

**Author contributions**

Research plan: X.B. Jin, C.L. Liu

Experimental design: X.B. Jin, C.L. Liu, A.J. Poulton

Financial support: C.L. Liu

Methodology (SEM, LM, statistical tools): A.J. Poulton

Experimental implementation and data analysis: X.B. Jin

Manuscript drafting and reviewing: X.B. Jin, A.J. Poulton

Nutrients and carbonate chemistry data support: M.H. Dai, X.H. Guo

**Acknowledgements**

This work is financed by the National Natural Science Foundation of China (grants 91228204, 41376047). I am grateful to the

cruise colleagues of R/V *Dongfanghong II* and Ocean Carbon Group of Xiamen University for their supplement of the nutrient and carbonate chemistry data. I am also grateful to R.B. Pearce, R.M. Sheward and G.M. Fragoso for their assistance in light and scanning electron microscopy, and H.E.K. Smith for her assistance in statistical analysis (National Oceanography Centre). I also thank M. Wang and H.R. Zhang for AVISO raw data compiling. A.J. Poulton would also like to acknowledge financial support from National Capability funding from the Natural Environmental Research Council.

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




**Tables**

**Table 1.** Sampling date, location, depth and upper water structure conditions: mixed layer depth (MLD), euphotic zone depth (Zeu).

| Station | Date (GMT+8) | Longitude | Latitude | Sampling depth (m) | MLD (m) | Zeu (m) |
|---------|--------------|-----------|----------|--------------------|---------|---------|
| D9 | 2014/6/25 7:11 | 119 | 18 | 25,50,75,100,150 | 34 | 95 |
| F1 | 2014/6/26 3:38 | 118 | 18 | 25,50,75,100,150 | 24 | 92 |
| G2 | 2014/6/26 14:36 | 117 | 18 | 25,45,75,100,150 | 12 | 90 |
| H3 | 2014/6/27 15:08 | 116 | 18 | 25,60,75,100,150 | 11 | 98 |
| I1 | 2014/6/20 0:52 | 115 | 19.5 | 25,50,100 | 16 | 76 |
| I2 | 2014/6/20 20:50 | 115 | 19 | 25,50,75,100,150 | 16 | 69 |
| I3 | 2014/6/29 9:23 | 115 | 18 | 25,50,75,100,150 | 23 | 99 |
| J1 | 2014/6/29 20:35 | 114 | 18 | 25,50,75,100,150 | 26 | 98 |
| X3 | 2014/6/30 6:58 | 113 | 18 | 25,50,75,100,150 | 30 | 100 |
| X4 | 2014/6/30 18:01 | 112 | 18 | 25,50,75,100,150 | 35 | 99 |
| X5 | 2014/7/1 5:10 | 111 | 18 | 25,50,75,100,150 | 17 | 93 |
| I4 | 2014/7/9 8:23 | 115 | 17 | 25,50,75,100,150 | 18 | |
| I5 | 2014/7/9 1:54 | 115 | 16 | 25,50,75,100,150 | (<25) | |
| I6 | 2014/7/8 17:53 | 115 | 15 | 25,50,75,100 | (>25) | |
| I7 | 2014/7/7 22:33 | 114.67 | 14 | 25,50,75,100,150 | 20 | |



**Table 2.** Coccolithophore species composition in Group 1, Group 2 and Group 3. R: mean relative abundance; F: occurrence frequency.

| | Group 1 | | Group 2 | | Group 3 | |
|---|---|---|---|---|---|---|
| | R | F | R | F | R | F |
| *Algirosphaera robasta* | 0.39 | 23.53 | 2.22 | 66.67 | **19.78** | **92.86** |
| *Florisphaera profunda* | 0.35 | 17.65 | 1.34 | 41.67 | **43.81** | **100.00** |
| *Gladiolithus flabellatus* | 0.00 | 0.00 | 0.00 | 0.00 | **1.66** | **60.71** |
| *Emiliania huxleyi* | **36.97** | **94.12** | **66.84** | **100.00** | 22.65 | 92.86 |
| *Gephyrocapsa oceanica* | 2.29 | 41.18 | **10.23** | **91.67** | 1.65 | 46.43 |
| *Gephyrocapsa ericsoni* | 6.20 | 52.94 | 6.20 | 50.00 | 2.61 | 32.14 |
| *Umbellosphaera irregularis* | **34.35** | **94.12** | 0.86 | 41.67 | 0.24 | 7.14 |
| *Umbellosphaera tenuis* | 2.14 | 47.06 | 0.10 | 16.67 | 0.00 | 0.00 |
| *Discosphaera tubifera* | **4.41** | **82.35** | 0.11 | 8.33 | 0.00 | 0.00 |
| *Rhabdosphaera clavigera* | 0.82 | 23.53 | 0.04 | 8.33 | 0.00 | 0.00 |
| *Calcidiscus leptoporus* | 0.82 | 17.65 | 1.53 | 58.33 | 0.96 | 35.71 |
| *Oolithotus fragilis* | 3.64 | 35.29 | **6.95** | **83.33** | 3.87 | 78.57 |
| *Helicosphaera carteri* | 1.05 | 58.82 | 0.21 | 25.00 | 0.03 | 3.57 |
| *Syracosphaera* spp. | **3.92** | **94.12** | 1.56 | 83.33 | 1.55 | 53.57 |
| *Umbilibcosphaera sibogae* | 0.45 | 17.65 | 0.71 | 33.33 | 0.22 | 14.29 |
| *Calciosolenia* spp. | 0.49 | 23.53 | 0.48 | 58.33 | 0.41 | 21.43 |
| *Michaelsarisa* spp. | 1.71 | 35.29 | 0.61 | 41.67 | 0.54 | 25.00 |



**Table 3.** Pearson's product-moment correlations between mean distal shield length (DSL) of *E. huxleyi*, nitrate+nitrite (N), phosphate (P), pH, total alkalinity ($A_T$) and $\Omega_C$ (n=29). Bold number denotes $p < 0.05$; Grey filling numbers denote $p < 0.01$.

|     | N | P | pH | $A_T$ | $\Omega_C$ | DSL |
|-----|-----|-----|-----|-----|-----|-----|
| N   |     |     |     |     |     |     |
| P   | 0.99527 |     |     |     |     |     |
| pH  | -0.80343 | -0.82774 |     |     |     |     |
| $A_T$ | 0.59234 | 0.56391 | -0.18852 |     |     |     |
| $\Omega_C$ | -0.89006 | -0.89833 | 0.87055 | -0.50845 |     |     |
| DSL | 0.60117 | 0.57968 | -0.52694 | 0.27468 | **-0.3956** |     |

35





**Table 4.** Principal component analysis for environmental parameters, PC 1 and PC 2 attribute for 78.87% and 16.76% of variance, respectively.

|  | PC 1 | PC 2 |
|---|---|---|
| nitrate+nitrite | 0.492 | 0.032 |
| phosphate | 0.493 | -0.012 |
| pH | -0.435 | 0.500 |
| $A_T$ | 0.304 | 0.857 |
| $\Omega_C$ | -0.480 | 0.110 |

**Figures**

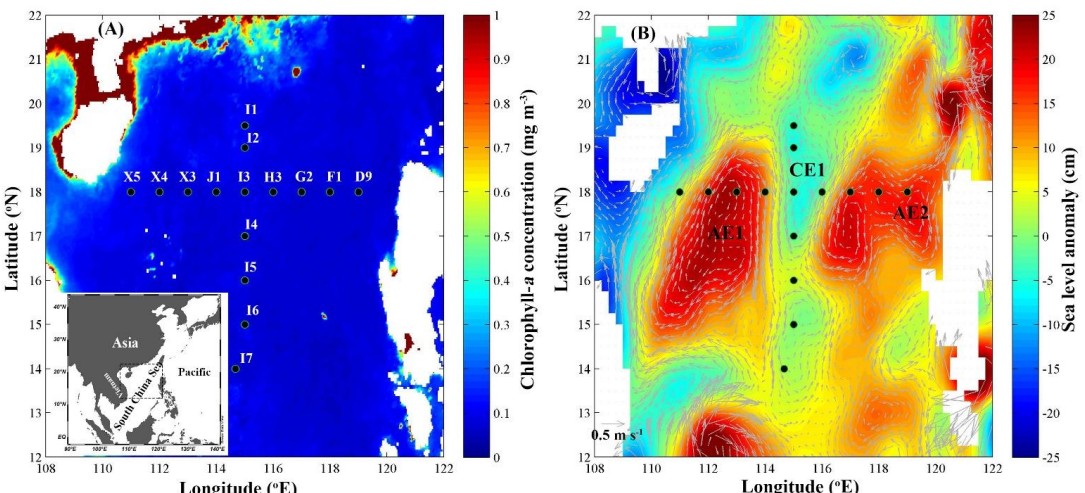

**Figure 1: (A) Sampling stations in the SCS, superimposed on the MODIS-Aqua (4 km) monthly average (May to August 2014) surface chlorophyll-*a* (mg m$^{-3}$). (B) Map of sea level anomaly (SLA) and surface flow in 30th June 2014. The positive SLA with clockwise flow indicates anti-cyclonic eddies (AE), and the negative SLA with anticlockwise flow indicates cyclonic eddies (CE).**



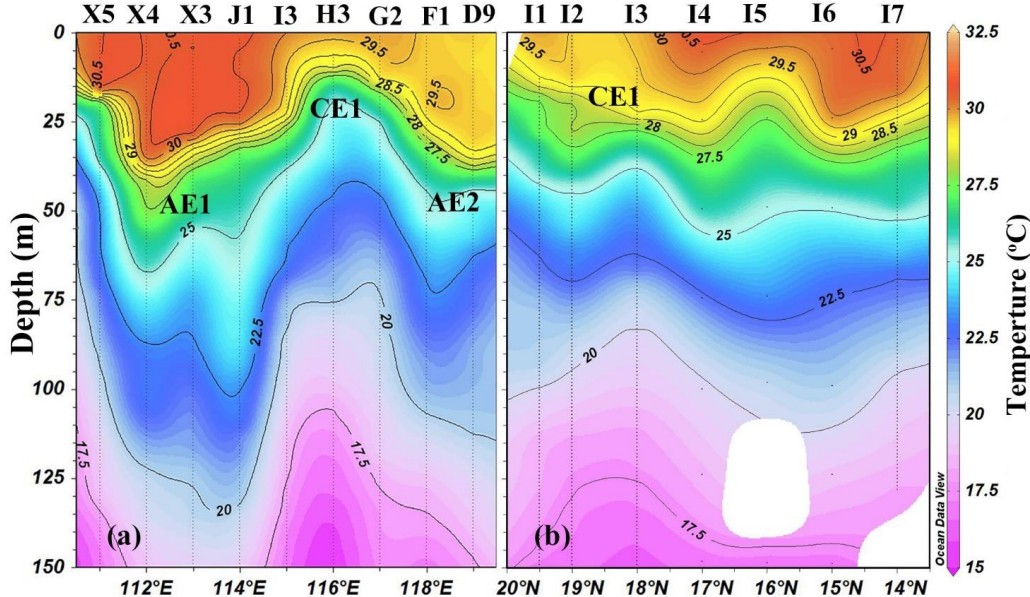

**Figure 2: Temperature ( ℃) profiles in zonal (a) and meridional (b) sections. Variation of isotherm indicates anti-cyclonic eddies (AE) and cyclonic eddy (CE) respectively. Profiles are dawn with Ocean Data View software (Schlitzer, 2015).**




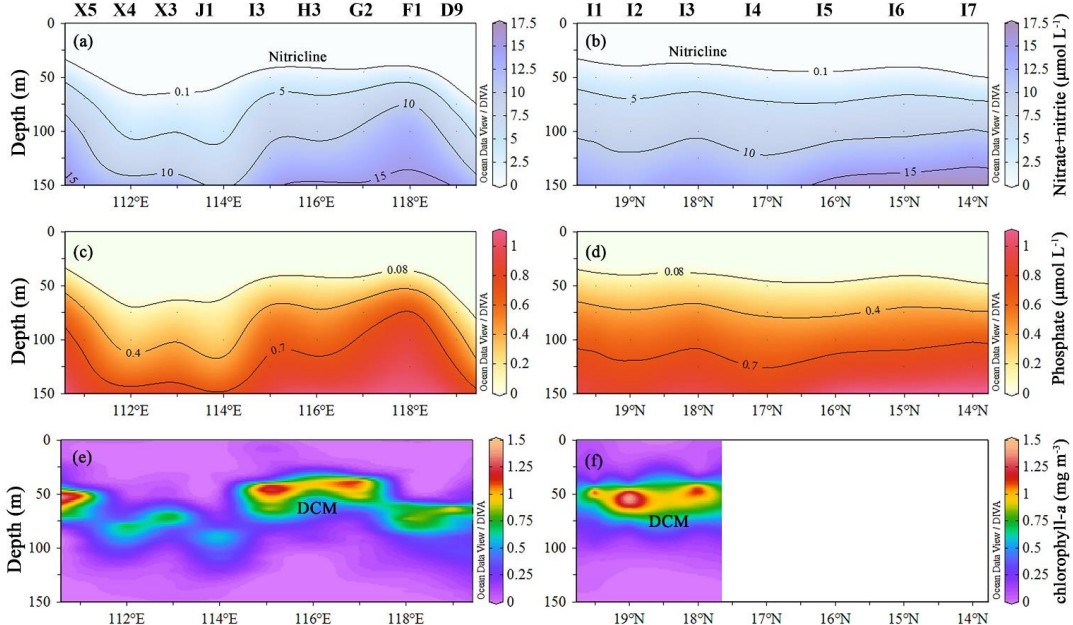

**Figure 3: Profiles of macronutrient (nitrate+nitrite, phosphate) condition and chlorophyll-*a* concentration (mg m⁻³) in zonal (a, c, e) and meridional sections (b, d, f). Nitricline is the depth where nitrate+nitrite is 0.1 µmol L⁻¹ (Borgne et al., 2002). DCM: deep chlorophyll-*a* maximum.**



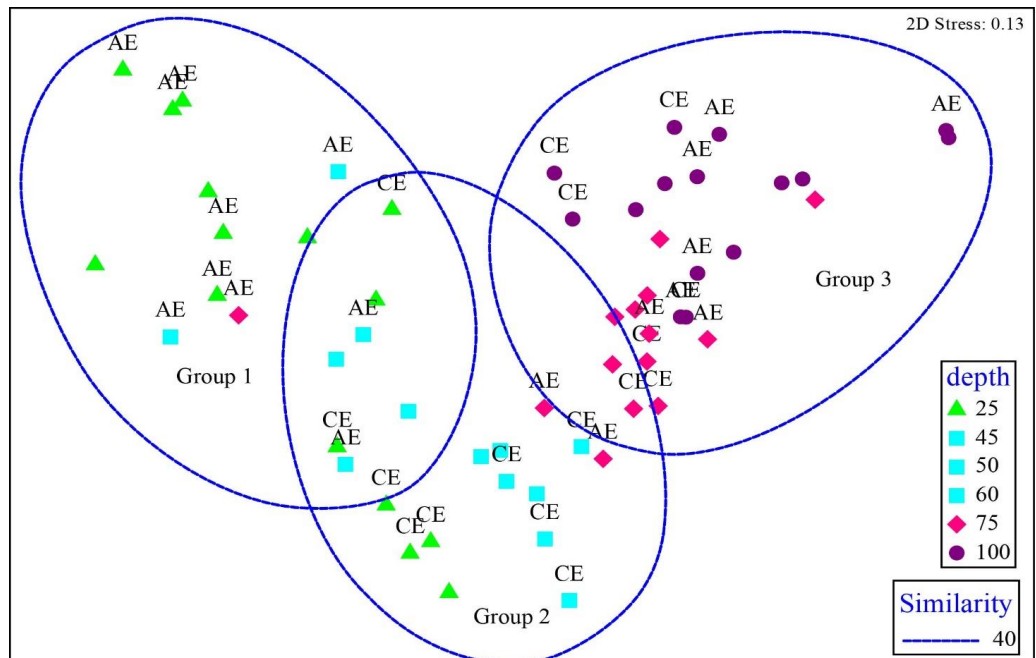

**Figure 4: Non-metric Multi-Dimensional Scaling (nMDS) ordination of stations in different depth, based on Bray-Curits similarity. The stress 0.13 of 2-dimentional ordination can provide a good interpretation for community group (Clarke and Warwick, 2001). The blue dashed lines indicate different divisions at 40 similarity, which is conducted by cluster analysis, using the same resemblance as nMDS. CE: cyclonic eddy; AE: anti-cyclonic eddy.**





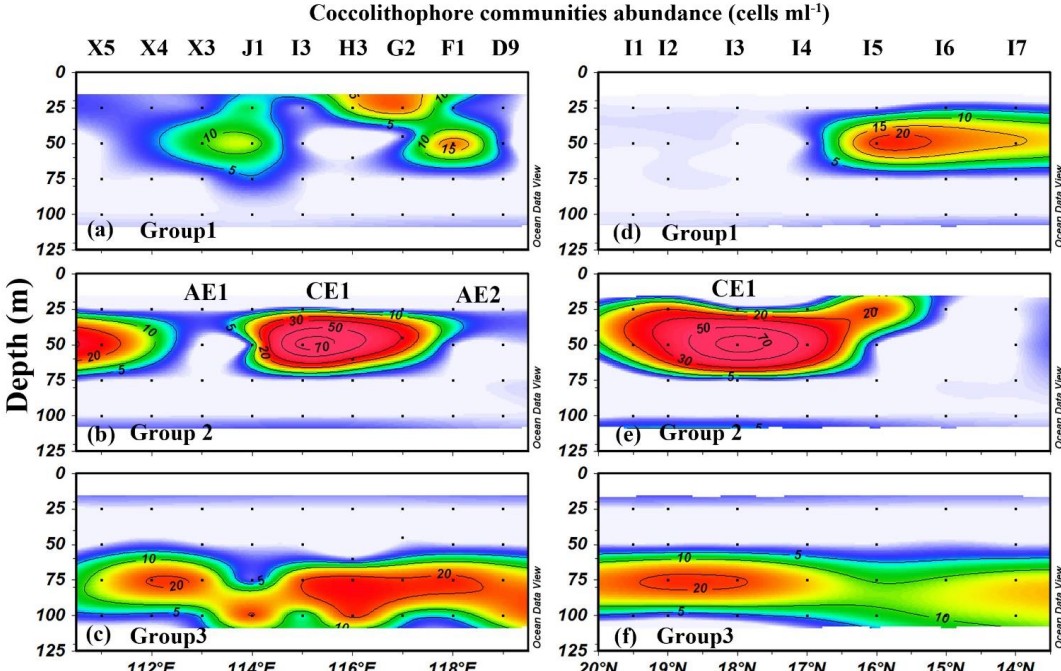

**Figure 5: Coccolithophore abundance (cells ml⁻¹) of three groups in zonal (a, b, c) and meridional sections (d, e, f). CE: cyclonic eddy; AE: anti-cyclonic eddy.**




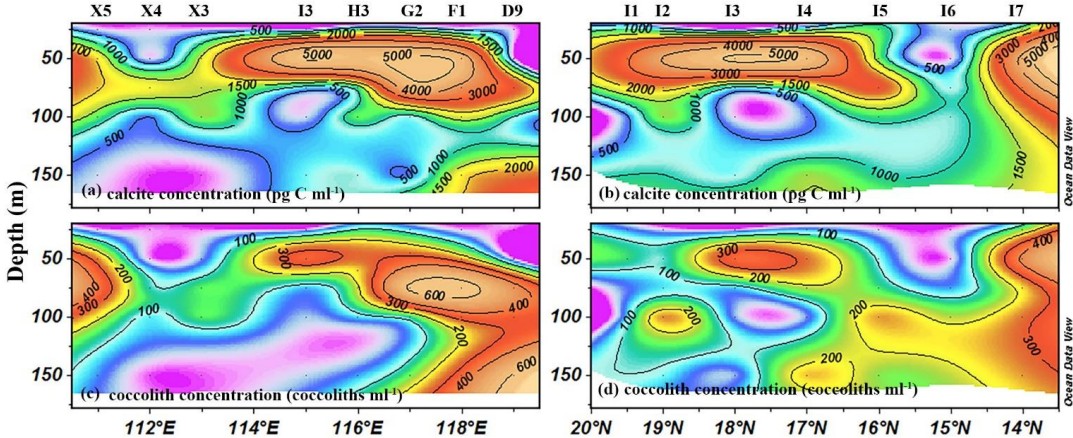

**Figure 6: Coccolithophore-based calcite concentration (a, b) and detached coccolith concentration (c, d) in zonal and meridional sections.**



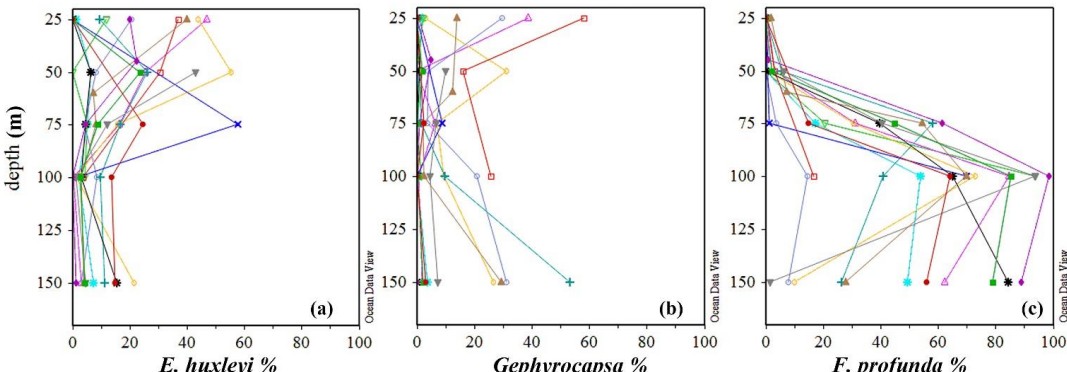

**Figure 7: The relative contribution of *E. huxleyi* (a), *Gephyrocapsa* spp. (b) and *F. profunda* (c) to total coccolithophore-based calcite concentration in water column.**




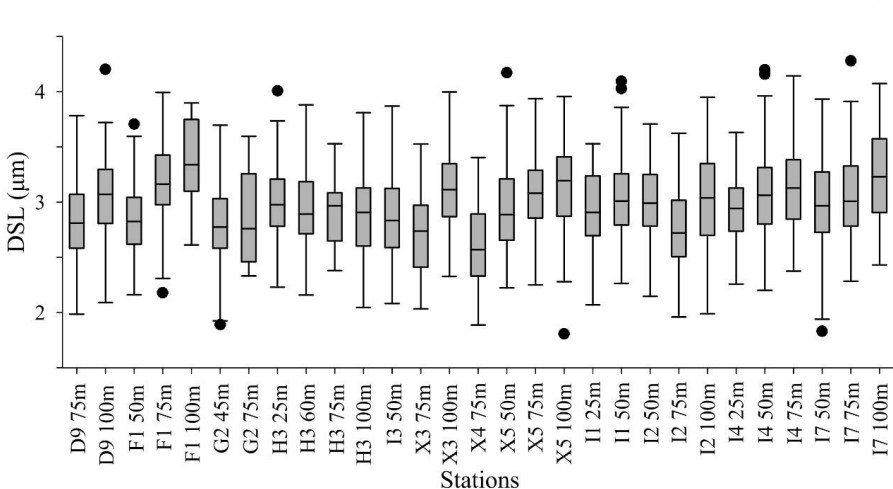

**Figure 8: Box plot of distal shield length (DSL) of *E. huxleyi* in measured stations shows median, minimum, maximum, upper and lower quartile and outliners.**



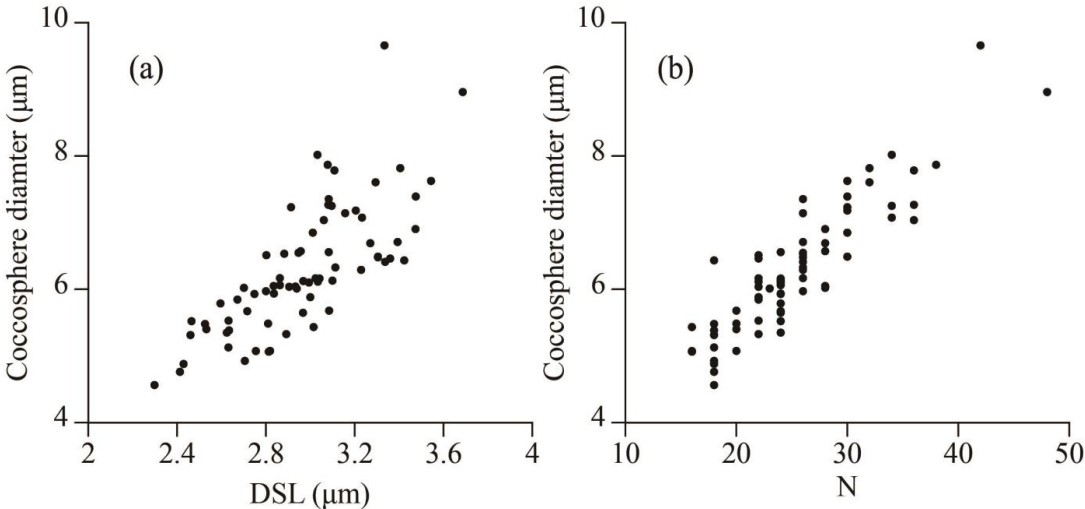

**Figure 9:** *E. huxleyi* **coccosphere diameter (CD) plotted versus distal shield length (DSL) (a) and coccolith number per coccosphere (N) (b), they both show good correlation. Coccosphere diameter can be estimated as: CD = 1.205 × DSL + 0.106 × N + 0.096.**



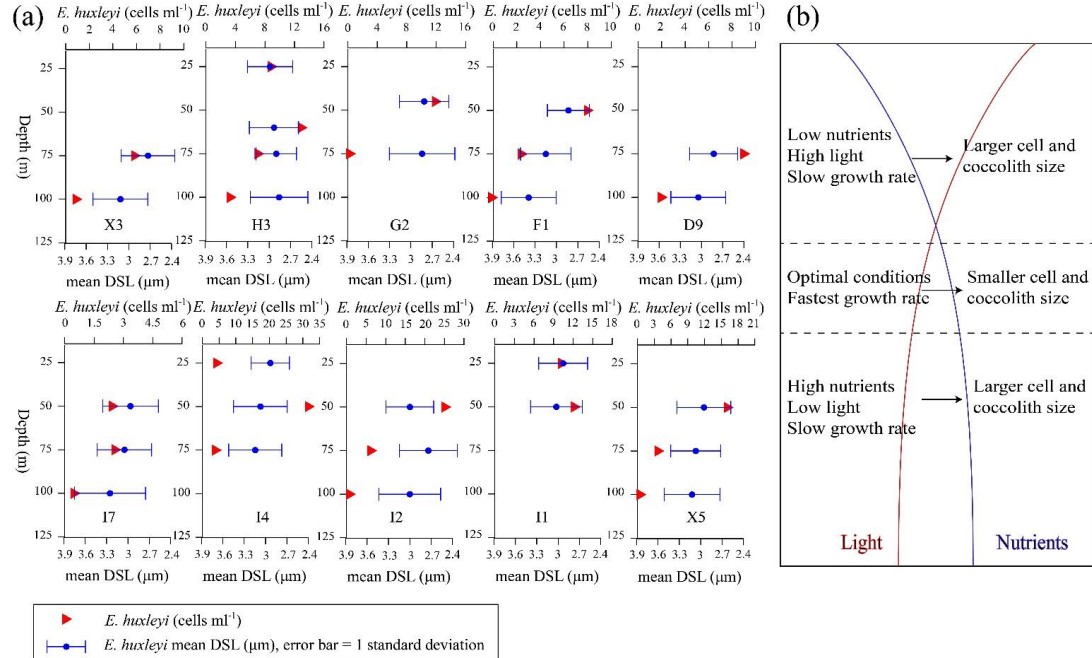

**Figure 10: (a) Cell abundance (red triangles) and mean distal shield length (DSL, blue dots, error bar =1 standard deviation) of *E. huxleyi* plotted in stations where there were at least two biometry measurement points. (b) A schematic map showing light and nutrients conditions in relation to coccolithophore growth rate and cell/coccolith size.**





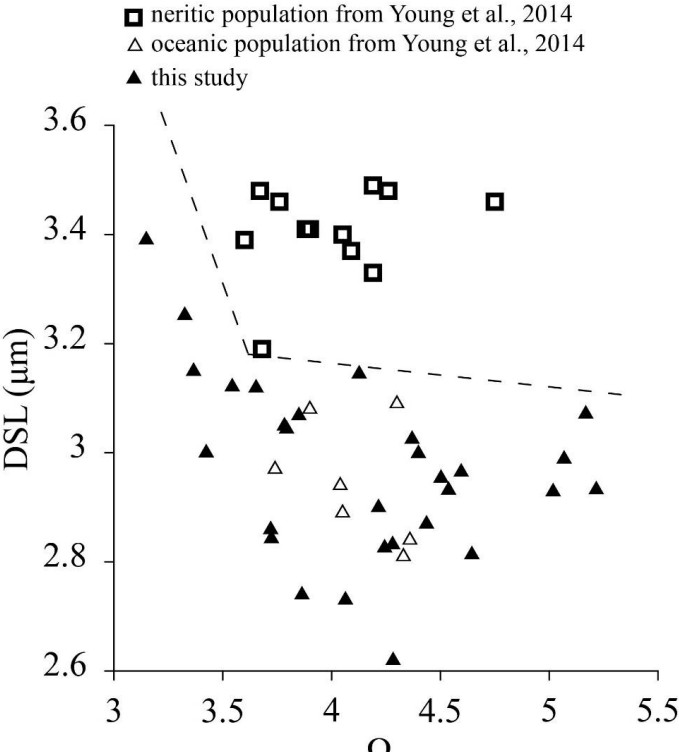

**Figure 11:** *E. huxleyi* **type A distal shield length (DSL) in the SCS (black triangles) with those in neritic (hollow triangles) and oceanic (hollow squares) in the North Sea (Young et al., 2014) plotted versus carbonate calcium saturation ($\Omega_C$).**



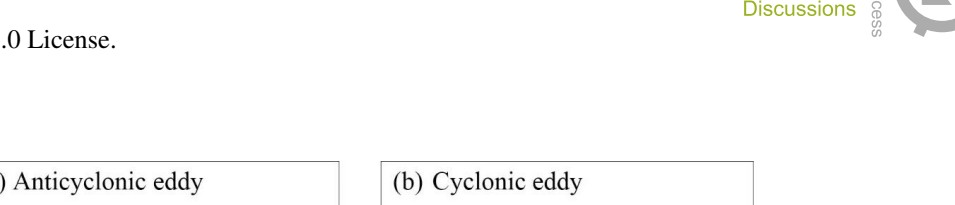

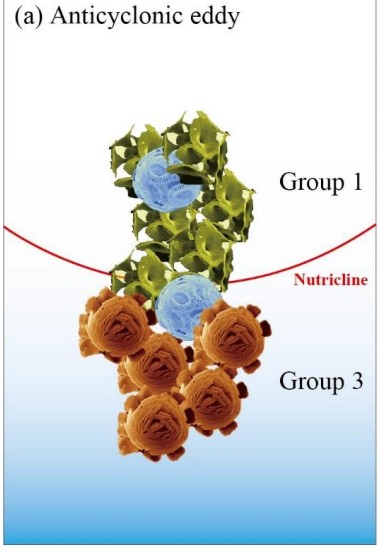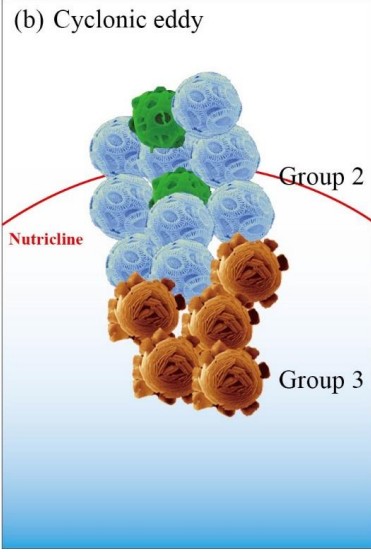

**Figure 12: Schematic of the coccolithophore communities in anticyclonic eddy (AE) and cyclonic eddy (CE). Communities sequence from UPZ to LPZ are Group 1 to Group 3 in AE (a) and Group 2 to Group 3 in CE (b), respectively.**

