# Peer review of "Coccolithophore responses to environmental variability in the South China Sea: species composition and calcite content"

_Biogeosciences, 2016_

## Referee Comment (RC1) · Anonymous Referee #1 · 4 Apr 2016

This manuscript shows composition of coccolithophores and contribution of each coccolithophore species/taxa to the calcite suspension in the water column in the South China Sea. Results from this study are useful for understanding of coccolithophore flora in the marginal sea. I would recommend publish this manuscript from the Biogeosciences after major revision. My comments are as follows;

Page 4 Line 1; 'Gladiolithus, Calciosolenia and Algirosphaera' are coccolithophore genus not coccolith species.

Page 6 Line 4, Page 8 Line 34, Page 8 Line 40; Three taxa not three species, since 'Gephyrocapsa spp.' includes multiple species.

Page 6 Line 6. and Figure 7; Authors mixed the coccoliths of Gephyrocapsa ericsonii and of Gephyrocapsa oceanica into a same category, Gephyrocapsa spp. in the estimation of calcite content, despite the volume/size of coccoliths of G. ericsonii is significantly smaller than that of G. oceanic. I would recommend authors to separate these two species from each other in the estimation of calcite content, revise Figure 7 with new estimation, and make discussion based on the new estimation.

Page 6 Lines 32-33; "The coccolithophore assemblages of the SCS are similar with those in the equatorial Pacific Ocean (Hagino et al., 2000)." Hagino et al. (2000) reported variation in coccolithophore assemblages in the equatorial Pacific. Which of the Hagino's assemblages resembles to the assemblage observed in this study?

Page 6 Lines 35-37; "However, in the equatorial and subtropical gyres of the Pacific and Atlantic Ocean, these coccolithophore species are found much deeper (150 m to 250 m) in the water column (Hagino et al., 2000; Boeckel and Baumann, 2008; Beaufort et al., 2008)." Hagino et al. (2000) studied coccolithophore assemblages in the equatorial upwelling front and in the Western Pacific Warm Pool, not in the gyre. By the way, what is the 'equatorial gyre'?

Page 7 Line 1; "Group 1 included umbelliform species, such as U. irregularis, which are considered K-selected (specialists) species" Please cite some papers that mentioned U. irregularis as K-selected species.

---

## Author Comment (AC1) · 12 Apr 2016

[This manuscript shows composition of coccolithophores and contribution of each coccolithophore species/taxa to the calcite suspension in the water column in the South China Sea. Results from this study are useful for understanding of coccolithophore flora in the marginal sea. I would recommend publish this manuscript from the Biogeosciences after major revision. My comments are as follows]

We would like to thank the reviewer for the helpful comments to improve the discussion paper.

[Page 4 Line 1; 'Gladiolithus, Calciosolenia and Algirosphaera' are coccolithophore

genus not coccolith species.]

Reply: 'species' has been revised as 'genera'.

[Page 6 Line 4, Page 8 Line 34, Page 8 Line 40; Three taxa not three species, since 'Gephyrocapsa spp.' includes multiple species.]

Reply: 'species' has been revised as 'taxa'.

[Page 6 Line 6. and Figure 7; Authors mixed the coccoliths of Gephyrocapsa ericsonii and of Gephyrocapsa oceanica into a same category, Gephyrocapsa spp. in the estimation of calcite content, despite the volume/size of coccoliths of G. ericsonii is significantly smaller than that of G. oceanica. I would recommend authors to separate these two species from each other in the estimation of calcite content, revise Figure 7 with new estimation, and make discussion based on the new estimation.]

Reply: Although in water column, the two species of Gephyrocapsa (G. oceanica and G. ericsonii) contributed similar cell population to Gephyrocapsa standing crops in coccolithophore ecological groups (Table 2), G. oceanica was the main contributor to Gephyrocapsa calcite inventory. For example G. oceanica and G. ericsonii comprised 7.00% and 0.87% (totally 7.87%, on average) of total calcite concentration, respectively. Figure 7 has been redrawn with estimation of these two species (Figure 7: The relative contribution of E. huxleyi (a), F. profunda (b), G. oceanica (c) and G. ericsonii (d) to total coccolithophore-based calcite concentration in water column. The black coarse lines denote moving average of 30 grid-points.). And we have supplemented the new estimation and discussion into relevant sections. (see red words in attached file)

[Page 6 Lines 32-33; "The coccolithophore assemblages of the SCS are similar with those in the equatorial Pacific Ocean (Hagino et al., 2000)." Hagino et al. (2000) reported variation in coccolithophore assemblages in the equatorial Pacific. Which of the Hagino's assemblages resembles to the assemblage observed in this study?]

Reply: We agree with the comment of the reviewer. In Hagino et al. (2000), coccolithophore florae are divided into four assemblages: High Temperature, Warm Oligotrophic, Warm Eutrophic and Temperature mixed-water. Coccolithophore taxa observed in the South China Sea resemble "High Temperature" and "Warm Oligotrophic" assemblages which include (ecological groups): UPG (U. irregularis, D. tubifera), LPG (F. profunda, A. robasta, G. flabellatus) and OPG (E. huxleyi). Hence, it has been rewritten as "The coccolithophore florae of the SCS are similar with 'High Temperature' and 'Warm Oligotrophic' assemblages in the equatorial Pacific Ocean (Hagino et al., 2000)."

[Page 6 Lines 35-37; "However, in the equatorial and subtropical gyres of the Pacific and Atlantic Ocean, these coccolithophore species are found much deeper (150 m to 250 m) in the water column (Hagino et al., 2000; Boeckel and Baumann, 2008; Beaufort et al., 2008)." Hagino et al. (2000) studied coccolithophore assemblages in the equatorial upwelling front and in the Western Pacific Warm Pool, not in the gyre. By the way, what is the 'equatorial gyre'?]

Reply: Apologizing for the vague description, here we meant coccolithophores in the West Pacific Warm Pool (stratified water, not mixed water and upwelling region) studied by Hagino et al. (2000), the subtropical gyre of the Pacific (by Beaufort et al., 2008) and of the Atlantic (by Boeckel and Baumann, 2008). The sentence has been rephrased that "However, in the West Pacifica Warm Pool (stratified waters) and subtropical gyres of the Pacific and Atlantic Ocean, species F. profunda are found much deeper (150 m to 250 m) in the water column (Hagino et al., 2000; Boeckel and Baumann, 2008; Beaufort et al., 2008)."

[Page 7 Line 1; "Group 1 included umbelliform species, such as U. irregularis, which are considered K-selected (specialists) species" Please cite some papers that mentioned U. irregularis as K-selected species.]

Reply: reference added: 'Young, 1994. Functions of coccoliths. In Coccolithophores,

edited by Winter, A. and Siesser, W.G., Cambridge University Press.'

Please also note the supplement to this comment:
http://www.biogeosciences-discuss.net/bg-2016-77/bg-2016-77-AC1-supplement.pdf

―――――――――――――――――――

[Figure]

**Fig. 1.** Figure 7

**Supplement:**

[revised manuscript text omitted]
 our investigation and the smaller (DSL<2 μm) and lightly calcified *G. ericsonii* is also the major component of *Gephyrocapsa* cells (Table 2); secondly, some coccolith sorting mechanisms (chemical or mechanical) should work between living cells and their fossil counterparts during sedimentation.

These taxa (*E. huxleyi, F. profunda* and *Gephyrocapsa* spp.) represented around half of the calcite in the water column (Fig. 7), whereas other species with smaller levels of abundance contribute to the other 50% of water column calcite. The importance of these relatively less abundant species in calcite inventories is partly related to higher per coccolith calcite contents, due in part to larger coccolith lengths and calcite contents (Young and Ziveri, 2000); for example, *Oolithus fragilis* has >80 pg C per coccolith whereas *E. huxleyi* has ~2 pg C per coccolith. Hence, relatively rare coccolithophore species with high coccolith and coccosphere calcite contents are important vectors of both upper-ocean calcite production (Daniels et al., 2014) and deep-sea calcite fluxes (Ziveri et al., 2007). However, in the case of the northern SCS basin, examination of sediment trap materials shows that the three taxa dominating upper ocean calcite inventories (*E. huxleyi, F. profunda, Gephyrocapsa* spp.) also represent >95% and >80% of deep-sea coccolith and coccolith-calcite fluxes (Jin et al., in prep.). This highlights the discrepancy of coccolith calcite species between euphotic zone and deep layers. Notably, at 150 m for some stations (D9, F1, G2, I5, X3), these three taxa can totally comprise more than 70% to 90% of calcite inventories and the contribution of *G. oceanica* exceed those of *E. huxleyi* (Fig. 7), which is similar with the cases of sediments in sea floor and mooring traps. It implies that coccolith sedimentation processes (coccolith sorting, aggregation) may primarily have taken place in the photic zone (e.g. upper ~100 m in the SCS). Within this zone, for example, a supra-lysoclinal dissolution of coccolith possibly occurred through grazing (Beaufort et al., 2007).

[revised manuscript text omitted]

---

## Referee Comment (RC2) · Anonymous Referee #2 · 26 Apr 2016

The manuscript by Jin et al. presents the results from a detailed taxonomical and ecological analysis of coccolithophores in the South China Sea. The latter may be of great use to broaden the knowledge of coccolithophore dynamics and to calibrate the use of coccoliths as environmental proxies. The sampling provided a complete dataset with a good vertical resolution containing just few gaps within stations. Given the importance of coccolithophores in both, the organic and in- organic carbon pumps, investigating their relationship with seawater carbonate chemistry is particularly relevant in our days. I see this study making a significant contribution to our understanding of coccolithophore distribution and responses to environmental variability, as it includes information on species composition and coccolith morphology against a broad set of

environmental parameters. However, the current version of the paper needs:

1. A clear separation between cyclonic and "normal" conditions and/or clearer methods to explain how it was done.

2. The effect of temperature and other parameters probably measured during the cruise (oxygen, salinity)

3. Parts of the discussion lack from clarity (please see specific comments)

4. Several figures are poorly discussed (i.e. figures 7, 8, 9).

Specific comments:

Abstract:

L13-15: "All living coccolithophores produced within. . . eddy centers" please check this sentence, it is long and difficult to understand.

Introduction:

P1.L24: "phytoplankton carbon"? Contribution to PIC is specified but not to POC

P1.L28: Consider ( up to 3 x 105 coccoliths ml-1, . . .)

P2.L7-23: This paragraph brings out a scientific question that is not answered in this work. The discussion about F. profunda distribution is rather a repetition of the statements presented in this part of the introduction. As this is not the central point of the manuscript and the raised question is not later answered, I suggest leaving the paragraph out of the introduction.

Methods

P3.L20: ". . .around 50 extra FOVs were examined" to reach a minimum of XX coccospheres.

P3.L24: How many coccoliths were counted?

P4.L3: Please give reference for the ks value of Florisphaera profunda (0.0016); in Young and Ziveri (2000) is ∼0.04.

P4.L22: "(Fig. 2), altimeter data on....and surface water flow.."

P4.L21-23: This is important for the discussion and I think it could be clearer. For instance, Fig. 2 is based on data from the 30.06.2014, but sampling took place from the 20.06 to the 09.07.2014. How was the calculation done for each sampled area? This may be important for the definition of your cyclonic eddy, which shows a SLA close to cero (from Fig. 2) and it might have been a "normal" condition like you assume for stations I1, I6 and I7.

Results

P6.L6: Please consider: water column coccolith calcite concentrations

P6.L11-12: Please consider moving into Methods section. Starting of the paragraph would be Emiliania huxleyi...

P6.L19-20: Please consider moving into Methods section. Starting of the paragraph would be: The mean coccosphere...

P6.L24-25: Were there any differences in coccolith size of morphotypes A and B? Was this somehow reflected in the distribution patterns and/or related to environmental parameters?

Discussion

P7.L14-22: How does this relates to the present study? I think it will be better to discuss the possible differences between E. huxleyi morphotypes A and B (as previously mentioned), how this does relates with temperature and then compare with the data you mention from previous works. Otherwise, the purpose of the whole paragraph is not very clear.

P7.L41: It is clear for surface samples but less clear for deeper samples. There were

only few samples from 75 m fitting in group 2 and they belonged to both cyclonic and anti-cyclonic eddies.

P8.L36: consider deleting "and calcite contents"

P8.L41-42: You meant that the contribution of the other (potentially larger) species decreased in the deep layers?

P9. Here it will be good to have a short discussion on the effects of temperature P9.L7: "..to change" how?

P9.L8: I am not sure it can be stated that malformed coccoliths have less calcite or even that they are smaller (which may be more relevant for this work). In fact, P limitation did not produce malformation of E. huxleyi coccoliths but it tended to increase the percentage of overcalcified (definition based in the spaces between distal shield elements) coccoliths and the coccosphere size (Oviedo et al. 2014). This without a clear pattern in PIC quotas. Also, Langer et al. (2011) reports no consistent correlation between coccolith morphology and growth or calcification rate.

P10.L1-7: This paragraph may be unnecessary because you already explained this "paradox" in page 9 L25 and 26, following Müller et al. (2008). In L5, what do you mean by "maturity under different limitation"?

P10.L28: Triantaphyllou et al. (2010) actually associates the seasonal variation with temperature rather than with nutrients, and thus, I think it would be good to check possible relations with temperature.

P10.L31-32: Please consider changing "assemblages" by "E. huxleyi populations"

P10.L32: "...but also in geological records" this conclusion cannot be extracted from the data presented in this work.

Conclusions

The second aim of the study (regarding potential paleo-ecological relationships) is not

reflected in the conclusions

Table 2. Please check species names

Figure 2. The same station I3 does not show the same structure in the two plots, were the measurements taken in different dates? Otherwise please explain.

Figure 7. Please add the legend for symbols-color codes.

Figures 7 and 8. Figures are poorly discussed. Could they go in supplementary material?
* * *
References Jeremy R. Young, Ziveri P (2000) Calculation of coccolith volume and its use in calibration of carbonate flux estimates. Deep Sea Res Part II 47:1679–1700 Langer G, Probert I, Nehrke G, Ziveri P (2011) The morphological response of Emiliania huxleyi to seawater carbonate chemistry changes: an inter-strain comparison. J Nannoplankton Res 32:29–34 Oviedo AM, Langer G, Ziveri P (2014) Effect of phosphorus limitation on coccolith morphology and element ratios in Mediterranean strains of the coccolithophore Emiliania huxleyi. J Exp Mar Biol Ecol 459:105–113 Triantaphyllou M, Dimiza M, Krasakopoulou E, Malinverno E, Lianou V, Souvermezoglou E (2010) Seasonal variation in Emiliania huxleyi coccolith morphology and calcification in the Aegean Sea (Eastern Mediterranean). Geobios 43:99–110
* * *

---

## Author Comment (AC2) · 9 May 2016

**Reply to Referee #2**

X.B. Jin

386jinxiaobo@tongji.edu.cn

The manuscript by Jin et al. presents the results from a detailed taxonomical and ecological analysis of coccolithophores in the South China Sea. The latter may be of great use to broaden the knowledge of coccolithophore dynamics and to calibrate the use of coccoliths as environmental proxies. The sampling provided a complete dataset with a good vertical resolution containing just few gaps within stations. Given the importance of coccolithophores in both, the organic and inorganic carbon pumps, investigating their relationship with seawater carbonate chemistry is particularly relevant in our days. I see this study making a significant contribution to our understanding of coccolithophore distribution and responses to environmental variability, as it includes information on species composition and coccolith morphology against a broad set of environmental parameters. However, the current version of the paper needs:

Thank you for your emphasizing on the importance of our works and the comments are helpful to improve this manuscript. Our responses are listed below.

1. A clear separation between cyclonic and "normal" conditions and/or clearer methods to explain how it was done.

[**About eddies**] Thanks, we have re-considered the coccolithophore communities with their relationship to hydrography, and rewritten some discussions in relevant sections.

As the Fig.1 shown, during sampling days of 18 °N section from 6-25 to 7-01, 2014, there was a stable eddy configuration in the South China Sea (SCS). So the Figure 2 in the manuscript paper can be representative for the eddy settings at least for 18 °N section. As there were two anti-cyclonic eddies (including st. X4, X3, JI, F1, D9) in the east and west part of the section, and between these anti-cyclonic eddies there was a cyclonic eddy (I3, H3), which could be identified by the negative SLA and the anti-clockwise surface flow. The cyclonic flow was most remarkable during 25 to 28 June, and weakened in 30, June. Hence, the "normal" conditions may include X5 and G2 station. The anti-cyclonic eddies can also be verified by the MLD variations in this section. For example, MLDs were deeper in X3 (30 m), X4 (35 m) and D9 (34 m), whereas MLD was shallowest in H3 (11 m). As the comments have referred, however, the cyclonic and normal conditions may not be recognized by temperature profiles clearly. Nevertheless, the eddy influences on coccolithophore groups' distribution were clear in this section. We have redrawn the Figure 5 in manuscript paper, that is the Fig.4 here, which has shown that in "normal" stations, all the coccolithophore groups occurred, as group 1 in ~25 m, group 2 in 50 m and group 3 within range of 75~100 m. In cyclonic eddy, group 2 was within the range from 25 to 50 m, and the coccolithophore abundances were highest in these groups. In anti-cyclonic eddies, there were two situations. One was that the range of group 1 was expanded, from 25 to 50 m. Locations of group 2 (75 m) were deeper, and group 3 were compressed in 100 m layer. Another was that group 2 was disappeared, and the coccolithophore maximum layer appeared in group 3, which was dominated by LPZ assemblages like *F. profunda*. DCMs were also deeper in anti-cyclonic eddies than in cyclonic and normal conditions, ranging from ~75 to 100 m.

Due the discontinuous sampling dates and low resolution of environmental data in some stations, the meridional section may not be suitable for accessing the eddy impacts on coccolithophore communities. For example, at I6 and I7 stations were not attributed for anti-cyclonic eddies from the SLA and surface flow map, however, the coccolithophore community locations are similar with those in anti-cyclonic eddies. This may be due to the intrinsic deeper nutricline in the center basin of SCS, even if water structure had not been modulated by eddies in sampling dates. Another example is for I1 and I2 stations, of which the coccolithophore groups were agreed with those in cyclonic eddies. Likewise, this was also not attributed for cyclonic eddies, as shown by SLA and surface flow. Interestingly, in I1 and I2 stations, the euphotic zone depth were relatively shallow (~70 m), as more light attenuation from suspended particles, which could be caused by the elevated particle production, since lots of diatom fragments were observed in SEM images. This finding corresponds with their station locations in the edge of anti-cyclonic eddy where particulate organic carbon (POC) fluxes can be 2 to 4 folder higher than those in adjacent oligotrophic waters (Zhou et al., 2013; Shih et al., 2015). The case for station I4 was similar with I1 and I2, as it located in the edge of two large anti-cyclonic eddies. The horizontal advection,

for water mass balance, can result in the elevated nutricline in the cyclonic-eddy edges, and hence the enhancement of POC production (Zhou et al., 2013).

[Figure]

5    Fig.1 SLA and surface flow allocations in the sampling dates of 18 °N section from 6-25 to 7-01, 2014.

[Figure]

Fig.2 SLA and surface flow in 2014-6-25, when station I1 and I2 were sampled.

[Figure]

Fig.3 SLA and surface flow during 7-9, July, 2014, when I4, I5, I6 and I7 were sampled.

[Figure]

[Figure]

Fig.4 Coccolithophore abundances and community group distribution in the investigation. LPZ included *F. profunda*, *A. robusta*, *G. flabellatus*; UPZ included *Umbellosphaera* spp., *D. tubifera*, *R. clavigera*.

2. The effect of temperature and other parameters probably measured during the cruise (oxygen, salinity)

[**About temperature**] Since, temperature has been mentioned for many times in the comments, we have briefly discussed it here to express our viewpoint that temperature may exert little impact on coccolithophore in the South China Sea (SCS).

Firstly, on coccolithophore communities:

Temperature should rule the coccolithophore biogeography in modern oceans. Winter et al. (1994) has demonstrated a meridional distribution of coccolithophore communities, from tropical, temperate to sub-polar zones. Nevertheless, our sampling was just confined to one season in the basin of SCS, which possess a stable sea surface temperature (SST), i.e. SST was more than 29 ℃ in all stations. Coccolithophore communities in our investigation exclusively belong to the "tropical" group. Temperature did change with depth, and the temperature gradient was ~ 0.1 ℃/m below the MLD. However, temperature co-varied (so did salinity and oxygen) with other environmental parameters, e.g. nutrients, light and carbonate chemistry, which may be more crucial. As coccolithophore groups, such as Umbelliform, Placolith and Floriform groups (Young, 1994), are more likely to link with nutrient and light rather than temperature. Actually, the predominant occurrence of *E. huxleyi* type A is related to the high SST in the SCS, as this morphotype is of warm water preference (Hagino et al., 2005) relative to type B and its derivatives.

Secondly, on *E. huxleyi* size:

Temperature did affect *E. huxleyi* size in a wide range (from 7 ℃ to 27 ℃) (Watabe and Wilbur, 1966). Within a relative small range (from 10 ℃ to 20 ℃), *E. huxleyi* size remained stable (Fielding et al., 2009). Considering the case in the SCS, the temperature difference from 50 m (~DCM) to 100 m is ~ 5 ℃, a rather small range. We suggest that coccolith size may be more insensitive to temperature in tropical seas than in high latitude regions, where the seasonality is more remarkable. Two different ways can account for how temperature lead to *E. huxleyi* size changes: physiologically and ecologically (Bach et al., 2012). The former is unlikely, as stated by Bach et al. (2012): "Temperature seems to have a small *physiological* influence on *E. huxleyi* coccolith size." A seasonal variation of *E. huxleyi* size was reported in the eastern Mediterranean Sea (Triantaphyllou et al., 2010). This may be due to overturning of different extent of calcifying *E. huxleyi* strains around one year (Triantaphyllou et al., 2010). However, this ecological effect is of minor possibility in the present case, as the relative small temperature range in summer in the basin SCS. At last, as discussed above, all parameters were changed with depth synchronously, making it hard to relate temperature to coccolith size.

3. Parts of the discussion lack from clarity (please see specific comments)

Responses are in specific comments.

4. Several figures are poorly discussed (i.e. figures 7, 8, 9).

Figure 7 is maybe important in discussion (re-discussed in section 4.3). Figure 8 and 9 have been moved into supplementary figures.

Specific comments:

Abstract:

L13-15: "All living coccolithophores produced within…eddy centers" please check this sentence, it is long and difficult to understand.

Thanks, this part of the abstract has been rewritten.

Introduction:

P1.L24: "phytoplankton carbon"? Contribution to PIC is specified but not to POC

"Phytoplankton carbon" here indicates organic carbon production from the reference (e.g. Poultion et al., 2010), so we rephrase it as primary production to avoid ambiguity.

P1.L28: Consider (up to $3 \times 10^5$ coccoliths ml$^{-1}$ …)

We have supplemented some information to this sentence: High concentrations of the cosmopolitan coccolithophore species *Emiliania huxleyi* generate large quantities of cells and detached coccoliths (e.g. ~2000 cells ml$^{-1}$ and $3\times10^5$ coccoliths ml$^{-1}$, Balch et al., 1991).

P2.L7-23: This paragraph brings out a scientific question that is not answered in this work. The discussion about F. profunda distribution is rather a repetition of the statements presented in this part of the introduction. As this is not the central point of the manuscript and the raised question is not later answered, I suggest leaving the paragraph out of the introduction.

Thanks, this paragraph has been cut out. Since, we have not answered the question about how *F. profunda* links to paleo-ecology and its difference from the published literatures.

Methods

P3.L20: "…around 50 extra FOVs were examined" to reach a minimum of XX coccospheres.

As the low abundance of coccolithophore in depth of 25 m and 150 m, around 150 FOVs were examined. For most cases about ~50 to 100 coccospheres were counted, still in some samples no coccolithophore was found. If N=1 and FOV=150 were given in the equation: C (ml$^{-1}$) = N $\times$ S / (A $\times$V), this would bring out a resolution of cell abundance as 0.09 cells ml$^{-1}$. We think this value is confident for coccolithophore abundance estimation in water-column.

P3.L24: How many coccoliths were counted?

These numbers are highly variable and to large extent dependent on the coccolith abundance for individual sample. Usually 250 to >400 detached coccoliths and ~10 to 20 coccospheres were counted in SEM images, however, in some barren samples, just <10 or no coccolith was found. The total 144 SEM images equals ~1.1 ml of seawater filtered.

P4.L3: Please give reference for the ks value of *Florisphaera profunda* (0.0016); in Young and Ziveri (2000) is ~0.04.

Sorry for mistake. Here, we assumed that single body *G. flabellatus* coccolith length and volume was 4 times larger than *F. profunda*, for their similar rectangle shapes. Single coccolith volume=$K_{GF}\times L_{GF}^3$=$5\times K_{FP}\times L_{FP}^3$, $L_{GF}$=5$\times L_{FP}$. So, $K_{GF}$=1/25$\times K_{FP}$=0.04/25=0.0016.

P4.L22: "(Fig. 2), altimeter data on...and surface water flow..."

P4.L21-23: This is important for the discussion and I think it could be clearer. For instance, Fig. 2 is based on data from the 30.06.2014, but sampling took place from the 20.06 to the 09.07.2014. How was the calculation done for each sampled area? This may be important for the definition of your cyclonic eddy, which shows a SLA close to zero (from Fig. 2) and it might have been a "normal" condition like you assume for stations I1, I6 and I7.

Please see [**About eddies**] above.

Results

P6.L6: Please consider: water column coccolith calcite concentrations

Specific coccolith calcite concentrations of *E. huxleyi*, *G. oceanica* and *F. profunda* in water column have been added.

P6.L11-12: Please consider moving into Methods section. Starting of the paragraph would be Emiliania huxleyi…

P6.L19-20: Please consider moving into Methods section. Starting of the paragraph would be: The mean coccospheres…

Thanks, these sentences have been moved into Methods section.

P6.L24-25: Were there any differences in coccolith size of morphotypes A and B? Was this somehow reflected in the distribution patterns and/or related to environmental parameters?

Sorry, detailed morphological measurements were just based on type A, which was the dominated morphotype for *E. huxleyi* in our investigation. Type B coccoliths were too few to make a confident biometry measurement as well as cells

counting for an individual sample from SEM images. Indeed, from the perspective of all samples (from SEM images), we can just make a conclusion that *E. huxleyi* A was the dominated morphotype. Their distribution patterns cannot be figured out, since cell counting for significant sample size was based on light microscope. However, we do believe that the general occurrence of these two morphotype is related to temperature.

Discussion

P7.L14-22: How does this relates to the present study? I think it will be better to discuss the possible differences between E. huxleyi morphotypes A and B (as previously mentioned), how this does relates with temperature and then compare with the data you mention from previous works. Otherwise, the purpose of the whole paragraph is not very clear.

10 This paragraph is intended to interpret why *E. huxleyi* type A dominated in the South China Sea. In the present work, regrettably, we have not a detail distribution pattern between type A and B which do not allow us discussing them in a more detailed way. However, we do believe that *E. huxleyi* type A is related to warm water mass, whereas type B prefers in cold water mass. As the southern South China Sea is one part of the West Pacific Warm Pool, in which annual average SST is higher than 28 ℃. Actually, from the temperature profile (Fig.2) SST was >28 ℃ even in a cold eddy. We have

15 rephrased this paragraph to stress the relationship between our observations with previous studies.

P7.L41: It is clear for surface samples but less clear for deeper samples. There were only few samples from 75 m fitting in group 2 and they belonged to both cyclonic and anti-cyclonic eddies.

The figure of cluster analysis (the same resemblance with nMDS) has been uploaded in supplementary figures to

20 illustrate the samples from group 1, group 2 and group 3. Indeed, group 2 fitting with 75 m (i.e. station J1) belonged to anti-cyclonic eddies.

P8.L36: consider deleting "and calcite contents"

Deleted.

P8.L41-42: You meant that the contribution of the other (potentially larger) species decreased in the deep layers?

Yes, there is a significantly loss of the calcite contribution of other species from photic layer (within 100 m) to deep ocean (≥500 m). New discussion has been made on this content.

30 P9. Here it will be good to have a short discussion on the effects of temperature P9. L7:"..to change" how?

Sorry, we do not believe that temperature is a key factor to influence *E. huxleyi* size in the present study. Please, see [**About temperature**].

P9.L8: I am not sure it can be stated that malformed coccoliths have less calcite or even that they are smaller (which may

35 be more relevant for this work). In fact, P limitation did not produce malformation of E. huxleyi coccoliths but it tended to increase the percentage of overcalcified (definition based in the spaces between distal shield elements) coccoliths and the coccosphere size (Oviedo et al. 2014). This without a clear pattern in PIC quotas. Also, Langer et al. (2011) reports no consistent correlation between coccolith morphology and growth or calcification rate.

We agree with the comments that the term "calcification" can mean three categories: (1) Malformation (calcification

40 abnormal), which is not necessarily related to coccolith calcite content, as reviewer mentioned. (2) PIC production (calcification rate), as estimated by equation: $\mu \times$ PIC/cell, which is to large extent dependent on growth rate. (3) Single coccolith mass (calcification for single coccolith), which is relied on coccolith morphological parameters (length, thickness, relative tube width). So, our results of coccolith DSL and their relationship with environmental conditions are belonging to the third conception and may possibly influence the calcification rate (the second conception) through

45 PIC/cell. And we have made a new discussion in this paragraph.

P10.L1-7: This paragraph may be unnecessary because you already explained this "paradox" in page 9 L25 and 26,

following Müller et al. (2008). In L5, what do you mean by "maturity under different limitation"?

In this paragraph, we have referred the traditional view of coccolithophore calcification and photosynthesis. As calcification can generate $CO_2$ (aq) for growth (photosynthesis), and photosynthesis can provide energy for calcification. We have supplemented the discussions about this traditional view and their differences with (e.g.) Müller et al. (2008), which is maybe important to understand this "paradox". Sorry, in L5, we meant that the different type of macro-nutrients needed for coccolithophore cell biomass growth and organic maturity (Aloisi, 2015). And this sentence has been moved to another section.

P10.L28: Triantaphyllou et al. (2010) actually associates the seasonal variation with temperature rather than with nutrients, and thus, I think it would be good to check possible relations with temperature.

Triantaphyllou et al. (2010) partly attributed the morphological change to ecophenotypic variation, apparently the seasonal changes may relate to temperature, since SST was ~26 ℃ in warm season, much higher than that in cold season (~14 ℃) in their study area. However, it is not the case in the South China Sea, which may be possibly due to two reasons. (1) Environmental parameters are highly co-varied including temperature. They are all depth dependent, as PCA shown. In addition, the seawater temperature variation is relatively small, i.e. temperature gradient is ~0.1 ℃/m below MLD. Combined with other parameters, we do not regard temperature as a key factor to influence coccolith size. (2) Our investigation is within one season in the SCS basin, a relatively small sampling range both for temporal and spatial. So, the ecophenotypic reason, possibly ruled by temperature, for coccolith size variation is unfavorable. Alternatively, it should be the *physiological* reason. However, some authors (Bach et al., 2012) stated that "*Temperature seems to have a small physiological influence on E. huxleyi size.*"

P10.L31-32: Please consider changing "assemblages" by "*E. huxleyi* populations"

P10.L32: "…but also in geological records" this conclusion cannot be extracted from the data presented in this work.

We have made a short discussion about wider implication not just limited to *E. huxleyi* populations in the present ocean, but also for the "ecological" influence on coccolith size variation in geological records.

Conclusions

The second aim of the study (regarding potential paleo-ecological relationships) is not reflected in the conclusions.

Thanks, the implication of some spices with their paleo-ecological relationships may not be summarized from our results, since our investigation was just limited to the oligotrophic summer season. So, this aim of the study has been removed.

Table 2. Please check species names

Thanks, corrected.

Figure 2. The same station I3 does not show the same structure in the two plots, were the measurements taken in different dates? Otherwise please explain.

Actually, this may result from the discontinuous sampling dates between I2, I3 and I4 stations, as I2 sampled at 6-20, I3 sampled at 6-29 and I4 sampled at 7-9 2014. Another reason may be the low resolution of temperature profile in I4 station, making it look slight difference due to the interpolation deviation in ODV plots.

Figure 7. Please add the legend for symbols-color codes.

Figure 7 has been redrawn.

Figures 7 and 8. Figures are poorly discussed. Could they go in supplementary material?

Figure 7 is maybe important in discussion (section 4.3). Figure 8 and 9 have been moved into supplementary figures.

[revised manuscript text omitted]
 observation, the predominant occurrences of morphotype A should be related to high temperature of seawater in the tropical SCS. As the southern part of SCS is also within the West Pacific Warm Pool, of which sea surface temperature is >28 ℃ annually. In general, *E. huxleyi* type A shows a warmer water preference than type B and other type B derivatives (C, B/C). For instance, *E. huxleyi* type A and type B dominated in the warm Kuroshio and cold Oyashio currents, respectively, off Japan (Hagino et al., 2005). In the Pacific sector of the Southern Ocean, *E. huxleyi* type A was found in the subantarctic zone, while type B, C and B/C were found in further south and colder waters (Saavedra-Pellitero et al., 2014).

*Lower photic zone (LPZ) assemblage:* In our study, the LPZ was represented by Group 3, which included typical LPZ species (*F. profunda*, *A. robusta* and *G. flabellatus*) and was found between 75 m and 100 m. Group 3 occurred above, at or near the depth where 1% of surface irradiance penetrated (i.e., base of the euphotic zone). In other tropical oceans, the LPZ assemblage dwells deeper than the base of the euphotic zone (Hagino et al., 2000; Boeckel and Baumann, 2008; Beaufort et al., 2008). In the northern Arabian Sea, *F. profunda* inhabits shallower waters, and is found across a wider depth range (10 m to 80 m) (Andruleit et al., 2003). It is worth noting that, as in the SCS, the Arabian Sea is strongly controlled by a monsoonal system (Indian monsoon) and is considered relatively eutrophic (Andruleit and Rogalla, 2002; Andruleit et al., 2003). Hence, it can be inferred that neither water depth or light availability are limiting factors for *F. profunda* (and/or other LPZ species) in the SCS, but rather nutrient availability is important; the nitricline is shallow (50 m to 75 m) even in the oligotrophic summer in the SCS.

**4.2 The response of coccolithophores to eddies in the South China Sea**

Mesoscale eddies have a strong influence on productivity and ecosystem structure in the SCS (Chen et al., 2007b; Lin et al., 2010; Zhang et al., 2011). Previous measurements in the SCS have shown that integrated primary production in cyclonic eddies can be 2-3 fold higher relative to the outside of eddies (Chen et al., 2007b). Modelling results have also highlighted how new production, relative to outside of eddies, can be ~30% higher or lower in cyclonic or anti-cyclonic eddies, respectively (Xiu and Chai, 2011).

With further examination of the nMDS, HCA and eddy settings in the 18 °N section, it clearly showed that the coccolithophore communities were strongly coupled with eddy occurrences (Fig. 5). In cyclonic eddy (I3, H3), the Group 2 occurred in ranges from 25 to 50 m depth and Group 3 occurred within layers from 75 to 100 m. Comparatively, in stations (X5, G2) as "normal" conditions, three groups sequentially occurred in water-column, as Group 1 in 25 m, Group 2 in 50 m and Group 3 in 75 to 100 m depth. In anti-cyclonic eddies, there were two situations: one was that Group 1 distributed within a wider range (from 25 to 50 m), and Group 3 was just within 100 m layer; another was that Group 2 was absent, and coccolithophore maximum layers were in Group 3, which were dominated by LPZ assemblages (e.g. *F. profunda*). This transition indicates the importance of ecology effects of eddies on coccolithophore community's

allocation in water-column. As the anti-cyclonic eddy (cyclonic eddy) centers are convergence (divergence) of the adjacent water, deepening (shallowing) the nutricline and making the water-column more oligotrophic (slightly eutrophic), of which conditions favor distinct coccolithophore assemblages (Fig. 8).

Due the discontinuous sampling dates (Table 1) and low resolution of environmental data in some stations, the meridional section may not be suitable for accessing the eddy impacts on coccolithophore communities. For example, at I6 and I7 stations were not attributed for anti-cyclonic eddies from the SLA and surface flow map, however, the coccolithophore community locations are similar with those in anti-cyclonic eddies. This may be due to the intrinsic deeper nutricline in the center basin of SCS, even if water structure had not been modulated by eddies in our investigation. Another example is for I1 and I2 stations, of which the coccolithophore groups were agreed with those in cyclonic eddies. Likewise, this was also not attributed for cyclonic eddies, as shown by SLA and surface flow (supplementary figures). Interestingly, in I1 and I2 stations, the euphotic zone depth were relatively shallow (~70 m), as more light attenuation from suspended particles, which could be caused by the elevated particle production, since a great number of diatom fragments had been observed in SEM images (e.g. at I2 25 m). This finding corresponds with their station locations in the edge of anti-cyclonic eddy where particulate organic carbon (POC) fluxes can be 2 to 4 folder higher than those in adjacent oligotrophic waters (Zhou et al., 2013; Shih et al., 2015). The case for station I4 was similar with I1 and I2, as it located in the edge of two large anti-cyclonic eddies (supplementary figures). The horizontal advection, for water mass balance, can result in the elevated nutricline in anti-cyclonic eddy edges, and hence, the enhancement of POC production (Zhou et al., 2013).

Station I5 had another distinctive arrangement of species assemblages which was opposite to that found at the other stations sampled (Fig. 5); Group 2 was found at 25 m while Group 1 was at 50 m. Examination of the temperature profile shows that the 29.5 $°C$ isotherm was shallow and domed, while the 22.5 $°C$ isotherm was pushed deeper into the water column (Fig. 2b). Filters collected at 25 m and 50 m from I5 also had lots of diatom fragments, and relatively elevated coccolithophore abundances (21.75 and 22.59 cells ml$^{-1}$ in 25 and 50 m, respectively). We suggest that this feature may represent a mode-water eddy, as described by McGillicuddy et al. (2007) in the northeast subtropical Atlantic Ocean. McGillicuddy et al. (2007) observed elevated phytoplankton production (i.e. a diatom bloom) in a mode-water eddy, which led to local changes in the zooplankton community composition (McGillicuddy et al., 2007; Eden et al., 2009).

**4.3 Calcite concentrations in the South China Sea**

The discrete estimates of bulk coccolith calcite roughly co-varied with coccolith and coccolithophore concentration in water column, with peak concentrations around the DCM. Rather than controlled by the environmental factors (light, nutrients, carbonate chemistry), the vertical distribution of bulk coccolith calcite reflected changes in the coccolithophore community composition. For example, the specific calcite contribution of *E. huxleyi* and *F. profunda* were the reflection of coccolithophore community changes in water-column. They contributed more in cyclonic eddy and less in anti-cyclonic eddies. In addition, excluding the maximum calcite concentration in the DCM, another peak was also found in deeper water at some stations, for example at 150 m in F1 and D9, and 100 m and 150 m in I7 station, where the living cells were low and calcite was nearly all contributed by detached coccoliths.

*E. huxleyi, G. oceanica* and *F. profunda* represented around half of the calcite in the water column (Fig. 7), whereas other species with smaller levels of abundance contribute to the other 50% of water column calcite. The great contribution of these relatively less abundant species in calcite inventories is partly related to higher per coccolith calcite contents, due in part to larger coccolith lengths (Young and Ziveri, 2000); for example, *O. fragilis* has >80 pg C per coccolith whereas *E. huxleyi* has ~2 pg C per coccolith. Relatively rare coccolithophore species with high coccolith and coccosphere calcite contents should be important vectors of both upper-ocean calcite production (Daniels et al., 2014) and deep-sea calcite fluxes (Ziveri et al., 2007). However, examination of sediment trap materials (500 m depth, 1500 m to sea floor) in the northern SCS basin shows that these three species (*E. huxleyi, G. oceanica* and *F. profunda*) dominating upper ocean calcite inventories have an increased contribution to coccolith (>95%) and coccolith calcite (>85%) fluxes (Jin et al., in prep.). This highlights the discrepancy of coccolith calcite species between euphotic surface

and aphotic deep oceans. Notably, at 150 m for some stations (D9, F1, G2, I5, X3), these three species can totally comprise more than 70% to 90% of calcite inventories and the contribution of *G. oceanica* exceed those of *E. huxleyi* (Fig. 7), which is similar with the cases in sediments of mooring traps. One possible reason is that these coccoliths are attributed for lateral transport of nepheloid layer originated from continental shelf or slope. This is most likely for D9 and F1 stations, as they have such high detached coccolith concentrations (Fig. 6) and located in the westernmost of 18 °N section. Alternatively, coccoliths in deep layer are result from vertical sinking. It indicates that the higher contribution of these species in deep layer may result from their higher production rate in photic zone which cannot be reflected from the snapshot-like discrete sampling.

**4.4 Environmental influences on Emiliania huxleyi biometry**

*Nutrients and light:* Some culture experiments have shown that nutrients may exert little influence on coccolith calcification rate or morphological variance (Paasche, 1998; Fritz, 1999; Langer and Benner, 2009; Langer et al., 2012). In mesocosm enclosures coccolith size has been shown to change under low phosphate conditions (Båtvik et al., 1997; Engel et al., 2005). A culturing study of *E. huxleyi* strains isolated from the Mediterranean Sea showed an increase in coccospheres size and cell calcite content under phosphorus limitation (Oviedo et al., 2014). These different results of *E. huxleyi* calcite quota or calcification rate under nutrient limitation may result from strain-specific responses (Oviedo et al., 2014). Until recently, a detail model has revealed that nitrogen and phosphorus are requisite for distinct cellular usage, biomass growth and organic maturity, respectively (Aloisi, 2015). Phosphorus deficient will delay cell division, whereas coccolithophore can still grow when nitrogen is not limited, so this mechanism has accounted for why cellular particulate inorganic carbon (PIC) increases under phosphorus stress (Aloisi, 2015). In the present study, a positive relationship between nutrients (both nitrogen and phosphorus) and *E. huxleyi* coccolith size was found (Table 3). Actually, the largest coccoliths occurred at deepest depth where nutrient was sufficient and light was insufficient, while within the *E. huxleyi* abundant layer coccoliths were relatively small (e.g. most remarkable at X3, F1, D9, I7, X5, Fig. 9a).

If nutrients are the only limiting factor in *E. huxleyi* growth (i.e. under culturing conditions), when nutrients are replete, *E. huxleyi* growth is fast (exponential phase), with fewer and smaller coccospheres per cell. When nutrients become limiting, *E. huxleyi* growth slows (stationary phase), and larger and multi-layer coccospheres are produced (Gibbs et al., 2013; O'Dea et al., 2014). A culturing experiment of *E. huxleyi* strain NIES 837 has shown that during rapid cell division phase, coccoliths production on cells was ceased (Satoh et al., 2009). In our cases in the SCS (in field conditions), nutrients were not the only limiting factor influencing *E. huxleyi* growth. We propose that light is also a limiting factor for *E. huxleyi* production and calcification in natural communities. Although some authors stated that light should not be regarded as a factor in phytoplankton growth in oligotrophic SCS, because the euphotic depth exceeds the MLD and nutricline throughout the year (Tseng et al., 2005; Wong et al., 2007). Here, a simple schematic is proposed (Fig. 9b): we suggest that nutrients are not limiting below the nutricline. (1) In the DCM layer, when light and nutrients are optimal for phytoplankton growth, *E. huxleyi* growth is fast and they produce small sized coccoliths; (2) In deeper waters, when nutrients are more sufficient but light is not available, *E. huxleyi* growth slows and they produce larger sized coccoliths. That light limitation, in *E. huxleyi* cell, can prolong G1 assimilation stage during which calcification take place will at last increase cellular calcite content (Müller et al., 2008); (3) Above the nutricline, when light is sufficient and nutrients are depleted, it is possible that *E. huxleyi* coccolith size is depended on whether phosphorus is deficient with reference to the Redfield ratio, although we have not enough data to support this contention as *E. huxleyi* coccoliths were too few in the SEM images in these samples for statistical significance.

The same trend of calcification in the water column has also been found off the Loffoten Islands in the Norwegian Sea (Charalampopoulou et al., 2011). Cell calcification rate was <1 pmol C cell$^{-1}$ d$^{-1}$ in the coccolithophore maximum layer, while it was about three times higher in upper and lower waters where coccolithophores were less abundant (Charalampopoulou et al., 2011), although bulk calcification in their study was influenced by light and coccolihophore species changes (Charalampopoulou et al., 2011). That opposite results were found in Benguela coastal upwelling system where coccospheres and coccoliths in the DCM (~17 m) were larger than those at 50 m depth could be due to different

bloom stage of *E. huxleyi* (Henderiks et al., 2012). Largest coccoliths/coccospheres were reported in late exponential growth stage (11th day) in mesocosm experiments (Engel et al., 2005). With a closer inspection, in their experiments phosphate was exhausted at the 11th day (<0.05 μmol L$^{-1}$), while nitrate was not below detection limit until 13th day (Engel et al., 2005). It means that phosphorus limitation regulated growth rate (decrease) with co-variation of cellular calcification (increase, negative response) (Müller et al., 2008; Aloisi, 2015). However, it is not the case in the SCS, because both nutrients were replete at deeper depth, and growth rate was, we suggested, limited by light availability. Other contrary results came from sediment traps which showed that heaviest coccoliths weight of *E. huxley*i was linked to primary productivity in bloom seasons in tropical Atlantic and the Mediterranean Sea (Beaufort et al., 2007; Meier et al., 2014b). Nevertheless, these changes may account for the seasonal overturn of heavily and lightly calcified *E. huxleyi* (possibly different morphotypes) (Triantaphyllou et al., 2010; Meier et al., 2014b).

Because, coccolithophore calcification is a strongly light-dependent process as stated in many literatures (e.g. Poulton et al., 2007; 2010; 2014). Apparently, it seems paradoxical that light constrains coccolithophore growth rates and promotes cellular calcification, since photosynthesis and calcification are coupled (Paasche, 2001; Rost and Reibesell, 2004). Considering that, calcification (photosynthesis) rate is estimated by the equation: $\mu \times$ PIC (POC) cell$^{-1}$, light or nutrients limitation definitely lower the growth rate $\mu$, more exactly the cell division rate as detected by cell counters in many culturing experiments. Sine calcification and photosynthesis rates are both influenced by $\mu$ (to large extent), they obviously show strongly coupled relation. However, the cellular PIC or POC content, the second factor of the equation, does increase when light or phosphorus is limited (Müller et al., 2008; Aloisi, 2015). Thus, this paradox may come from the different perspective from coccolith calcification (rate). Overall, cell/coccolith size variations are a combination of physiological responses to environmental constraints, and may also be influenced by zooplankton grazing in natural conditions (Beaufort et al., 2007). Whether this response is positive or negative needs more detailed studies.

[revised manuscript text omitted]

Water samples were divided into 3 groups according to their coccolithophore communities. Group 1, represented by the presence of *U. irregularis*, preferred to oligotrophic conditions; Group 2, dominated by *E. huxleyi*, had relative high coccolithophore cells; and Group 3 contained lower photic assemblages. These coccolithophore communities in water-column showed strong vertical diffentiation, with response to mesoscale eddy features in 
[revised manuscript text omitted]

**Table 4.** Principal component (PC) analysis and Pearson product moment correlations (r) between environmental parameters (N: nitrate+nitrite, P: phosphate, pH, $A_T$, $\Omega_C$) and PC scores. PC-1 and PC-2 attribute for 78.87% and 16.76% of variance, respectively. **p<0.01, *p<0.05.

|  | PC-1 loading | PC-2 loading | PC-1 scores | PC-2 scores |
|---|---|---|---|---|
| N | 0.49 | 0.03 | 0.97** | 0.03 |
| P | 0.49 | -0.01 | 0.98** | -0.01 |
| pH | -0.43 | 0.50 | -0.86** | 0.45* |
| $A_T$ | 0.30 | 0.85 | 0.60** | 0.78** |
| $\Omega_C$ | -0.48 | 0.11 | -0.95** | 0.10 |

**Figure 1:**

[Figure]

**Figure 1: (A) Sampling stations in the SCS, superimposed on the MODIS-Aqua (4 km) monthly average (May to August 2014) surface chlorophyll-*a* (mg m⁻³). (B) Map of sea level anomaly (SLA) and surface flow in 30th June 2014. The positive SLA with clockwise flow indicates anti-cyclonic eddies (AE), and the negative SLA with anticlockwise flow indicates cyclonic eddies (CE).**

**Figure 2:**

[Figure]

**Figure 2: Temperature (°C) profiles in zonal (a) and meridional (b) sections. Variation of isotherm indicates anticyclonic eddies (AE) and cyclonic eddy (CE) respectively. Profiles are dawn with Ocean Data View software (Schlitzer, 2015).**

**Figure 3:**

[Figure]

**Figure 3: Profiles of macronutrient (nitrate+nitrite, phosphate) condition and chlorophyll-*a* concentration (mg m⁻³) in zonal (a, c, e) and meridional sections (b, d, f). Nitricline is the depth where nitrate+nitrite is 0.1 µmol L⁻¹ (Borgne et al., 2002). DCM: deep chlorophyll-*a* maximum.**

**Figure 4:**

[Figure]

**Figure 4: Non-metric Multi-Dimensional Scaling (nMDS) ordination of stations in different depth, based on Bray-Curits similarity. The stress 0.13 of 2-dimentional ordination can provide a good interpretation for community group (Clarke and Warwick, 2001). The blue dashed lines indicate different divisions at 40 similarity, which is conducted by cluster analysis, using the same resemblance as nMDS. CE: cyclonic eddy; AE: anti-cyclonic eddy.**

**Figure 5:**

[Figure]

**Figure 5: Coccolithophore abundance (cells ml⁻¹) of three groups in sampling stations. "LPZ" specifically indicates three species: *F. profunda*, *A. robusta* and *Gladiolithus flabellatus*. "UPZ" specifically indicates: *Umbellosphaera* spp.**

5 **(mainly *U. irregularis*), *D. tubifera* and *R. clavigera*.**

**Figure 6:**

[Figure]

**Figure 6: Coccolithophore-based calcite concentration (a) and detached coccolith concentration (b) in zonal and meridional sections.**

**Figure 7:**

[Figure]

**Figure 7: The relative contribution of *E. huxleyi* (a), *F. profunda* (b), *G. oceanica* (c) and their total contribution (d) to coccolithophore-based calcite concentration in water column. The black lines denote moving average of 30 grid-points.**

**Figure 8:**

[Figure]

**Figure 8: Schematic showing the coccolithophore communities in anti-cyclonic eddy and cyclonic eddy.**

**Figure 9:**

[Figure]

**Figure 9: (a) Cell abundance (red triangles) and mean distal shield length (DSL, blue dots, error bar =1 standard deviation) of *E. huxleyi* plotted in stations where there were at least two biometry measurement points. (b) A schematic map showing light and nutrients conditions in relation to coccolithophore growth rate and cell/coccolith size.**

**Figure 10:**

[Figure]

**Figure 10:** *E. huxleyi* **type A distal shield length (DSL) in the SCS (black triangles) with those in neritic (hollow triangles)**

**and oceanic (hollow squares) in the North Sea (Young et al., 2014) plotted versus carbonate calcium saturation ($\Omega_C$).**

---

## Author Response (AR1)

**5 **Reply to Referee #1**

[This manuscript shows composition of coccolithophores and contribution of each coccolithophore species/taxa to the calcite suspension in the water column in the South China Sea. Results from this study are useful for understanding of coccolithophore flora in the marginal sea. I would recommend publish this manuscript from the Biogeosciences after major revision. My comments are as follows]

10 Reply: We would like to thank the reviewer for the helpful comments to improve the discussion paper.

[Page 4 Line 1; 'Gladiolithus, Calciosolenia and Algirosphaera' are coccolithophore genus not coccolith species.] Reply: Thanks, revised.

15 [Page 6 Line 4, Page 8 Line 34, Page 8 Line 40; Three taxa not three species, since 'Gephyrocapsa spp.' includes multiple species.]

Reply: Thanks, these sections are rewritten.

[Page 6 Line 6. and Figure 7; Authors mixed the coccoliths of Gephyrocapsa ericsonii and of Gephyrocapsa oceanica into a same category, Gephyrocapsa spp. in the estimation of calcite content, despite the volume/size of coccoliths of G. ericsonii is significantly smaller than that of G. oceanica. I would recommend authors to separate these two species from each other in the estimation of calcite content, revise Figure 7 with new estimation, and make discussion based on the new estimation.]

Reply: Figure 7 has been redrawn. And the discussion has been rewritten into relevant sections.

25

40

[Page 6 Lines 32-33; "The coccolithophore assemblages of the SCS are similar with those in the equatorial Pacific Ocean (Hagino et al., 2000)." Hagino et al. (2000) reported variation in coccolithophore assemblages in the equatorial Pacific. Which of the Hagino's assemblages resembles to the assemblage observed in this study?]

- Reply: We apologize to the reviewer for the confusion. To clarify, in Hagino et al. (2000), the coccolithophore florae
  are divided into four assemblages: High Temperature; Warm Oligotrophic; Warm Eutrophic; and Temperature mixed-water. Coccolithophore taxa observed in the South China Sea resemble "High Temperature" and "Warm Oligotrophic" assemblages which include (the ecological groups): Upper Photic-zone Group (*U. irregularis, D. tubifera*), Lower Photic-zone Group (*F. profunda, A. robusta, G. flabellatus*) and Omnipresent Group (*E. huxleyi*). Hence, this line has been rewritten as "The coccolithophore florae of the SCS are similar, with 'High Temperature' and
- 35 'Warm Oligotrophic' assemblages, to those found in the equatorial Pacific Ocean (Hagino et al., 2000)."

[Page 6 Lines 35-37; "However, in the equatorial and subtropical gyres of the Pacific and Atlantic Ocean, these coccolithophore species are found much deeper (150 m to 250 m) in the water column (Hagino et al., 2000; Boeckel and Baumann, 2008; Beaufort et al., 2008)." Hagino et al. (2000) studied coccolithophore assemblages in the equatorial upwelling front and in the Western Pacific Warm Pool, not in the gyre. By the way, what is the 'equatorial gyre'?]

Reply: We apologize for the vague description, here we meant to refer to coccolithophores in the West Pacific Warm Pool (stratified water, not mixed water or an upwelling region) as studied by Hagino et al. (2000), the subtropical gyre of the Pacific (Beaufort et al., 2008) and of the Atlantic (Boeckel and Baumann, 2008). The sentence has now been

45 rephrased as "However, in the West Pacific Warm Pool (stratified waters) and subtropical gyres of the Pacific and Atlantic Ocean, species such as *F. profunda* are found much deeper (150 m to 250 m) in the water column (Hagino et al., 2000; Boeckel and Baumann, 2008; Beaufort et al., 2008)."

[Page 7 Line 1; "Group 1 included umbelliform species, such as U. irregularis, which are considered K-selected (specialists) species" Please cite some papers that mentioned U. irregularis as K-selected species.]

Reply: Reference has now been added to Young (1994) Functions of coccoliths.

**Reply to Referee #2**

The manuscript by Jin et al. presents the results from a detailed taxonomical and ecological analysis of coccolithophores in the South China Sea. The latter may be of great use to broaden the knowledge of coccolithophore dynamics and to calibrate the use of coccoliths as environmental proxies. The sampling provided a complete dataset

- 10 with a good vertical resolution containing just few gaps within stations. Given the importance of coccolithophores in both, the organic and inorganic carbon pumps, investigating their relationship with seawater carbonate chemistry is particularly relevant in our days. I see this study making a significant contribution to our understanding of coccolithophore distribution and responses to environmental variability, as it includes information on species composition and coccolith morphology against a broad set of environmental parameters. However, the current version of the paper needs:
  - Reply: Thank you for your emphasizing on the importance of our works and providing helpful comments to improve this manuscript. Our responses are listed below.

**1. A clear separation between cyclonic and "normal" conditions and/or clearer methods to explain how it was done.**

20 Reply: [About eddies] Thanks, we have now re-considered the coccolithophore communities and their relationship to hydrography, and rewritten some of the discussion in relevant sections.

As shown in figure 1 in supplementary materials, during sampling days of the 18 N section from 25 June to 1 July there was a stable eddy configuration in the South China Sea (SCS). Hence, the Figure 2 in the manuscript can be representative of the eddy settings, at least for the 18 N section. There were two anti-cyclonic eddies (including st. X4,

- X3, JI, F1, D9) in the east and west part of the section, and between these anti-cyclonic eddies there was a cyclonic eddy (I3, H3), which could be identified by the negative SLA and the anti-clockwise geostrophic flow. The cyclonic flow was most remarkable during 25 to 28 June, and weakened around 30 June. Hence, the "normal" conditions may include X5 and G2 stations. The presence of the anti-cyclonic eddies can also be verified by the MLD variations in this section. For example, MLDs were deeper in X3 (30 m), X4 (35 m) and D9 (34 m), whereas MLD was shallowest in
- 30 H3 (11 m). As the reviewer rightly points out, however, the cyclonic and normal conditions may not be clearly recognized by temperature profiles. Nevertheless, the eddy influences on coccolithophore distributions were clear in this section. We have redrawn the Figure 5 in the manuscript, which shows that at "normal" stations, all the coccolithophore floral groups occurred; group 1, ~25 m; group 2 at 50 m; and group 3 within the depth range of 75 to 100 m. In the cyclonic eddy, group 2 was within the depth range from 25 to 50 m, and this group had the highest
- 35 coccolithophore abundance. In the anti-cyclonic eddies, there were two different patterns. The first was that the depth range of group 1 was expanded from 25 to 50 m. The depth location of group 2 (75 m) was deeper, while group 3 was compressed in the 100 m layer. The second pattern was for group 2 to disappear, while group 3 had highest abundances, and was dominated by LPZ species like *F. profunda*. The DCMs were also deeper in the anti-cyclonic eddies than in the cyclonic eddies and in normal conditions, ranging in depth from ~75 to 100 m.

40

Due to the discontinuous sampling dates and low (depth) resolution of environmental data at some stations, the meridional section may not be completely suitable for accessing the eddy impacts on coccolithophore communities. For example, stations I6 and I7 were not classified to anti-cyclonic eddies from the SLA and geostrophic flow map, however the vertical structure and taxonomic composition of the coccolithophore community were similar to those in

45 the anti-cyclonic eddies. This may be due to the deeper nutricline in the central basin of the SCS, even if the water-column structure had not been modulated by eddies during the sampling dates. Another example is from stations I1 and I2, for which the coccolithophore distribution and species groupings agreed with those in the cyclonic eddies. Likewise, these stations were not classified as within cyclonic eddies, as shown by SLA and geostrophic flow. Interestingly, at the I1 and I2 stations, the euphotic zone depth was relatively shallow (~70 m), as more light was attenuated from suspended particles, which may have been caused by elevated particle production. This finding

corresponds with these station locations within the edge of an anti-cyclonic eddy where particulate organic carbon

- 5 (POC) fluxes can be 2 to 4 folder higher than those in the adjacent oligotrophic waters (Zhou et al., 2013; Shih et al., 2015). The case for station I4 was similar to I1 and I2, as it was located in the edge of two large anti-cyclonic eddies. The horizontal advection, for water mass balance, can result in the elevated nutricline in the cyclonic-eddy edges, and hence the enhancement of POC production and export (Zhou et al., 2013).
- 10 2. The effect of temperature and other parameters probably measured during the cruise (oxygen, salinity)

Reply: [About temperature] since temperature is mentioned several times in the comments, we will now briefly discuss it here to express our viewpoint that temperature may exert little impact on coccolithophores in the South China Sea (SCS). We have also added a new discussion about temperature influence on coccolith size to section 4.4 of the revised paper.

15

35

**Firstly, temperature and coccolithophore communities:**

At the oceanic basin-scale, temperature aligns with coccolithophore biogeography in the modern ocean: Winter et al. (1994) demonstrated clear meridional distribution of coccolithophore communities, from tropical, temperate to sub-polar zones. Nevertheless, our sampling was confined to one season in the SCS, which possess relatively stable sea
surface temperatures (SST), i.e. SST was more than 29 ℃ at all stations. Coccolithophore communities in our investigation exclusively belong to the "tropical" floral group. Temperature did change with depth, with a temperature gradient of ~ 0.1 ℃/m below the MLD. However, temperature co-varied (as did salinity and oxygen) with all other environmental parameters vertically, e.g. nutrients, light and carbonate chemistry, which may be more crucial in controlling growth rates and species composition. The different coccolithophore groups (such as Umbelliform, Placolith and Floriform groups; Young, 1994), are also more likely to favour different nutrient and light conditions, rather than temperature. Notably, the predominant occurrence of *E. huxleyi* type A is related to the high SST in the SCS, as this morphotype has a warm water preference (Hagino et al., 2005) relative to type B and its derivatives (Poulton et

al., 2011).

**30 Secondly, temperature and *E. huxleyi* coccolith size:**

Temperature does affect *E. huxleyi* coccolith size over a wide range (e.g., from 7  $^{\circ}$ C to 27  $^{\circ}$ C; Watabe and Wilbur, 1966). Within a relative small range (from 10  $^{\circ}$ C to 20  $^{\circ}$ C), *E. huxleyi* coccolith size has been found to remain fairly stable (Fielding et al., 2009). In the case the SCS, the temperature difference from 50 m (~DCM) to 100 m is ~ 5  $^{\circ}$ C, a rather small range. We suggest that coccolith size may be more insensitive to temperature in tropical seas than in high latitude regions, where seasonal changes in SST are stronger. Two different ways could an account for how

- temperature influences *E. huxleyi* coccolith size: physiologically and ecologically (Bach et al., 2012). The former is unlikely, as stated by Bach et al. (2012): "Temperature seems to have a small physiological influence on *E. huxleyi* coccolith size." A seasonal variation of *E. huxleyi* coccolith size was reported in the eastern Mediterranean Sea (Triantaphyllou et al., 2010). This may be due to overturning and succession between different strains *E. huxleyi* with
- 40 different extents of calcite content across the year (Triantaphyllou et al., 2010). However, this ecological effect is of minor possibility in the present case, as the relative small temperature range in summer in the SCS. Finally, as discussed above, all parameters changed with depth synchronously, making it hard to relate coccolith size to temperature alone.
- 45 3. Parts of the discussion lack from clarity (please see specific comments) Reply: Please see replies in specific comments.

**4. Several figures are poorly discussed (i.e. figures 7, 8, 9).**

Reply: Figure 7 is important in the discussion and is now re-discussed in section 4.3. Figures 8 and 9 have now been moved into the supplementary material.

**5 Specific comments:**

Abstract:

L13-15: "All living coccolithophores produced within...eddy centers" please check this sentence, it is long and difficult to understand.

Reply: Thanks, this part of the abstract has been rewritten.

**10**

**Introduction:**

**P1.L24: "phytoplankton carbon"? Contribution to PIC is specified but not to POC**

Reply: "Phytoplankton carbon" here indicates organic carbon production from the reference (e.g. Poulton et al., 2010), so we have rephrased it as primary production to avoid ambiguity.

15

**P1.L28: Consider (up to $3 \times 10^5$ coccoliths ml-1 ...)**

Reply: We have supplemented some information to this sentence: High concentrations of the cosmopolitan coccolithophore species *Emiliania huxleyi* generate large quantities of cells and detached coccoliths (e.g. ~2000 cells  $ml^{-1}$  and  $3 \times 10^5$  coccoliths  $ml^{-1}$ , Balch et al., 1991).

20

P2.L7-23: This paragraph brings out a scientific question that is not answered in this work. The discussion about F. profunda distribution is rather a repetition of the statements presented in this part of the introduction. As this is not the central point of the manuscript and the raised question is not later answered, I suggest leaving the paragraph out of the introduction.

25 Reply: Thanks, this paragraph has now been cut out.

**Methods**

**P3.L20: "...around 50 extra FOVs were examined" to reach a minimum of XX coccospheres.**

Reply: As the low abundance of coccolithophores at depths of 25 m and 150 m, around 150 FOVs were examined. For 30 most cases about ~50 to 100 coccospheres were counted, though in some samples no coccolithophore cells were found. 31 If N=1 and FOV=150 were given in the equation:  $C (ml^{-1}) = N \times S / (A \times V)$ , this would give a detection limit of 0.09 32 cells ml-1. We use this value as the lower limit of coccolithophore abundance estimated in the water-column.

**P3.L24: How many coccoliths were counted?**

- 35 Reply: These numbers are highly variable and to large extent dependent on the coccolith abundance for individual sample. Usually 250 to >400 detached coccoliths and ~10 to 20 coccospheres were counted in SEM images, however, in some barren samples,
- 40 P4.L3: Please give reference for the ks value of *Florisphaera profunda* (0.0016); in Young and Ziveri (2000) is ~0.04. Reply: Sorry for the mistake. Here, we assumed that single body *G. flabellatus* coccolith length and volume was 4 times larger than *F. profunda*, based on their similar rectangle shapes. Single coccolith volume =  $K_{GF} \times L_{GF}^3 = 5 \times K_{FP} \times L_{FP}^3$ ,  $L_{GF} = 5 \times L_{FP}$ . So,  $K_{GF} = 1/25 \times K_{FP} = 0.04/25 = 0.0016$ .
- 45 P4.L22: "(Fig. 2), altimeter data on...and surface water flow..."

P4.L21-23: This is important for the discussion and I think it could be clearer. For instance, Fig. 2 is based on data from the 30.06.2014, but sampling took place from the 20.06 to the 09.07.2014. How was the calculation done for each sampled area? This may be important for the definition of your cyclonic eddy, which shows a SLA close to zero (from Fig. 2) and it might have been a "normal" condition like you assume for stations I1, I6 and I7.

50 Reply: Please see [About eddies] above.

**5 Results**

P6.L6: Please consider: water column coccolith calcite concentrations

Reply: Specific coccolith calcite concentrations of *E. huxleyi*, *G. oceanica* and *F. profunda* in the water-column have been added.

10 P6.L11-12: Please consider moving into Methods section. Starting of the paragraph would be Emiliania huxleyi...

P6.L19-20: Please consider moving into Methods section. Starting of the paragraph would be: The mean coccospheres...

Reply: Thanks, these sentences have been moved into the Methods section.

15 P6.L24-25: Were there any differences in coccolith size of morphotypes A and B? Was this somehow reflected in the distribution patterns and/or related to environmental parameters?

Reply: Sorry, detailed morphological measurements were only based on type A, which was the dominant morphotype of *E. huxleyi* in our investigation. Type B coccoliths were too few to make any confident biometric measurements. Indeed, from the perspective of all samples (from SEM images), we can only make a conclusion that *E. huxleyi* A was

20 the dominant morphotype. Their distribution patterns cannot be determined as cell counts for significant sample sizes were based on light microscope – where the morphometric differences cannot be determined. However, we do believe that the general occurrence of these two morphotypes is partially related to temperature.

Discussion

25 P7.L14-22: How does this relates to the present study? I think it will be better to discuss the possible differences between E. huxleyi morphotypes A and B (as previously mentioned), how this does relates with temperature and then compare with the data you mention from previous works. Otherwise, the purpose of the whole paragraph is not very clear.

Reply: This paragraph is intended to interpret why E. huxleyi type A dominated in the South China Sea. In the present

- 30 work, regrettably, we have not a detail distribution pattern between type A and B which does not allow us to discuss them in a more detailed way. However, we do believe that *E. huxleyi* type A is related to warm water, whereas type B prefers cold water. The southern South China Sea is one part of the West Pacific Warm Pool, in which annual average SST is higher than 28 ℃ (even the the temperature profile (Fig. 2) was >28 ℃ in the cold water eddy). We have now rephrased this paragraph to stress the relationship between our observations and previous studies.
- 35

40

P7.L41: It is clear for surface samples but less clear for deeper samples. There were only few samples from 75 m fitting in group 2 and they belonged to both cyclonic and anti-cyclonic eddies.

Reply: The figure of cluster analysis (the same resemblance with nMDS) has been uploaded in the supplementary material to illustrate the samples from group 1, group 2 and group 3. Indeed, group 2 fitting with 75 m (i.e. station J1) belonged to the anti-cyclonic eddies.

P8.L36: consider deleting "and calcite contents" Reply: Deleted.

P8.L41-42: You meant that the contribution of the other (potentially larger) species decreased in the deep layers?
 Reply: Yes, there is a significantly loss of the calcite contribution of other species from photic layer (within upper 100 m) to deep ocean (deeper than 500 m). New discussion has been made on this content.

**P9. Here it will be good to have a short discussion on the effects of temperature P9. L7:"..to change" how?**

50 Reply: We do not believe that temperature is a key factor to influence *E. huxleyi* coccolith size in the present study. Please see [**About temperature**].

10

20

P9.L8: I am not sure it can be stated that malformed coccoliths have less calcite or even that they are smaller (which may be more relevant for this work). In fact, P limitation did not produce malformation of E. huxleyi coccoliths but it tended to increase the percentage of overcalcified (definition based in the spaces between distal shield elements) coccoliths and the coccosphere size (Oviedo et al. 2014). This without a clear pattern in PIC quotas. Also, Langer et al. (2011) reports no consistent correlation between coccolith morphology and growth or calcification rate.

- Reply: We agree with the comments that the term "calcification" can mean three categories: (1) Malformation (calcification abnormal), which is not necessarily related to coccolith calcite content, as reviewer mentioned. (2) PIC production (calcification rate), as estimated by equation: μ × PIC/cell, which is to large extent dependent on growth rate. (3) Single coccolith mass (calcification for single coccolith), which is determined by coccolith morphological parameters (length, thickness, relative tube width). So, our results of coccolith DSL and their relationship with
- environmental conditions belong to the third category and may possibly influence the calcification rate (the second conception) through PIC/cell. We have now made a new discussion in this paragraph.

**P10.L1-7: This paragraph may be unnecessary because you already explained this "paradox" in page 9 L25 and 26, following Müller et al. (2008). In L5, what do you mean by "maturity under different limitation"?**

- Reply: In this paragraph, we have referred to the traditional view of coccolithophore calcification and photosynthesis calcification can generate CO2 (aq) for growth (photosynthesis), and photosynthesis can provide energy for calcification. We have supplemented the discussion about this traditional view and their differences with (e.g.) M üller et al. (2008), which is maybe important to understand this "paradox". In L5, we meant to allude to the different type of macro-nutrients needed for coccolithophore cell biomass growth and organic maturity (Aloisi, 2015). This sentence
- 25 macro-nutrients needed for coccolithophore cell biomass growth and organic maturity (Aloisi, 2015). This sentence has been moved to another section of the paper.

**P10.L28: Triantaphyllou et al. (2010) actually associates the seasonal variation with temperature rather than with nutrients, and thus, I think it would be good to check possible relations with temperature.**

- 30 Reply: Triantaphyllou et al. (2010) partly attributed the morphological change to ecophenotypic variation, apparently the seasonal changes may relate to temperature, since SST was ~26 °C in warm season, much higher than that in cold season (~14 °C) in their study area. However, it is not the case in the South China Sea, which may be possibly due to two reasons. (1) Environmental parameters highly co-varied with temperature and are all depth dependent, as the PCA shows. In addition, the seawater temperature variation is relatively small, i.e. temperature gradient is ~0.1 °C/m below
- MLD. Combined with other parameters, we do not regard temperature as a key factor influencing coccolith size. (2) Our investigation is within one season in the SCS basin, a relatively small sampling range both temporally and spatially. Hence, the ecophenotypic reason, possibly ruled by temperature, for coccolith size variation is unfavorable. Still, we have made a new discussion about temperature's influence on coccolith size in section 4.4.

**40 P10.L31-32: Please consider changing "assemblages" by "*E. huxleyi* populations"**

P10.L32: "...but also in geological records" this conclusion cannot be extracted from the data presented in this work. Reply: We have made a short discussion about wider implications, not just limited to *E. huxleyi* populations in the present ocean, but also for the "ecological" influence on coccolith size variation in geological records.

45 Conclusions

The second aim of the study (regarding potential paleo-ecological relationships) is not reflected in the conclusions.

Reply: Thanks, the implication of some species with their paleo-ecological relationships may not be summarized from our results, since our investigation was just limited to the oligotrophic summer season. This aim of the study has been removed.

50

Table 2. Please check species names

**5 Reply: Thanks, corrected.**

Figure 2. The same station I3 does not show the same structure in the two plots, were the measurements taken in different dates? Otherwise please explain.

Reply: Actually, this may result from the discontinuous sampling dates between I2, I3 and I4 stations, as I2 was sampled on 6-20, I3 sampled on 6-29 and I4 sampled on 7-9.

Figure 7. Please add the legend for symbols-color codes. Reply: Figure 7 has been redrawn.

15 Figures 7 and 8. Figures are poorly discussed. Could they go in supplementary material? Reply: Figure 7 is important in the discussion (section 4.3). Figures 8 and 9 have been moved into supplementary figures.

[revised manuscript text omitted]

---

## Referee Report (RR1)

**General comments**

In most cases, replies to comments satisfactorily treated the general issues raised. However the discussion requires revision in the form. In some cases the changes in the manuscript discussion made it even more confusing (some examples are: P16 L42-45; P18 L14-19). Because of this lack of clarity in the discussion I believe the manuscript is not ready to be published.

According to the MS title, species composition and calcite content are the key parameters under study, but the largest part of the discussion goes to *Emiliania huxleyi* biometry (already in the introduction is E. *huxleyi* biometry and not calcite content the main parameters). The discussion of this section (4.4) goes back and forth around the same ideas and should be rewritten so it gets more concise.

Regarding the figures, I think the changes improved the MS.

In general, I believe the data set, calculations and statistical work do deserve publication but the discussion has to be improved. Readability should still be improved.

English is not my mother language so I did not focus on it, but a revision of grammar will be useful.

**Specific comments**

Abstract:
L15. Perhaps some words on how eddies regulated the coccolithophore community?

Introduction: I agree with the changes.

Methods:
P11, L22 and 23: Not necessary here.
L28: "In addition to" instead of "Apart from" or just starting the sentence in "Coccolith length"
P12, L34: Similarity matrix was constructed with biomass data? It was not explained how biomass was estimated.

Results:
P13, L35: "these features may indicate lateral transport" instead of "these features may be characteristics of either lateral transport". Perhaps the whole sentence should go to the discussion.
P14, L9: Italics for species name

Discussion:
P15, L15: I do agree it was probably related to the higher temperatures in the sampled area; but insist that this is not proved with the data of this study.
L38: Sentence needs to be re-written, I suggest: Results from nMDS, HCA and eddy settings in the 18ºN section clearly showed that…..
P16, L29-31: thus, community composition did not change with environmental factors? In the previous section the differences between the results in cyclonic eddies, anti-cyclonic eddies, and adjacent stations were suggested to be driven by nutrients.

Same lines: It would be nice to elaborate more on the fact that *E. huxleyi* and *F. profunda* contributed more to water column calcite in cyclonic eddies than in anti-ciclonic eddies. This is not mentioned in the results section.
L42-45: I don´t understand "in anti-cyclonic eddies,… coccolithophore maximum layers were in group 3" does that mean that some stations in these eddies had a "coccolithophore maximum layer" while other stations did not? And that the first clustered together? Perhaps it refers to abundance instead of layer?
L45: "This transition" it should be clearer stated in the text

P16, L7: "coccolithophore community locations are similar with those in anti-ciclonic eddies" I don´t understand; does it means: environmental conditions or coccolithophores community composition/structure in those stations (I6 and I7) resembled those of AC eddies?.
L16-18: The last sentence of the paragraph is maybe not necessary as it was mentioned just before that POC fluxes are higher in the border of AC eddies.
L45: "coccolith calcite species" I understand what is meant but the term is confusing. It may be better stated as in the comment of the review and the response to this comment (P6, L26-28 in the Reply to referee #2)
P17, L1: please delete "totally"

P17, L9: I believe the whole section 4.4 should be re-written. The changes done so far did not help the MS. In this section, the argument that is proposed to explain the results stresses the importance of light limitation because nutrient limitation does not explain their results. However: 1) It is very difficult to follow the ideas that cut, come back, are exposed using confusing terms. 2) It goes back and forth around the same ideas: trying to explain why under P-limiting conditions cellular PIC generally has increased in culture experiments, although this is not "the core" of their arguments and could just be developed in one short paragraph. 3) PIC was not determined in this study but rather coccolith-coccosphere sizes and, although these parameters are obviously related, the author passes from PIC to coccolith size and to coccosphere size as equal without warning the reader.
L12: "coccoliths size has been show to change under low phosphate. .." how? Also, Engel et al. (2005) has been cited here and again in L2 of page 18 with the same purpose, can these paragraphs be synthesized into one?
L15: E. *huxleyi* calcite quota did not differ between the 6 strains tested in that experiment. The calcification rate did. Meaning growth rate changed in a strain specific fashion.
L16-17: This is not a novelty and it was not "revealed" by Aloisi´s model.
L17: "Phosphorus deficiency" instead of "phosphorus deficient". However, the whole sentence should be reconsider because Aloisi (2015) himself refered to Müller et al. (2008) (and others, please see page 4676). The same issue is raised again in page 18, the newly inserted L14-19. None of the two paragraphs (in page 17 or in page 18) are clearly written.

P18, L5: Müller et al. (2008), and Aloisi (2015) did not say that Engel´s et al. results meant that P limitation regulated growth rate…. This is the authors' interpretation.
L14 -19: It would be better to focous on PIC only and not to mention POC, since the whole discussion should go around coccolith biometry. Thus, parentheses are not necessary.
L19: "different perspective from coccolith calcification (rate)"?
L30: "possible" instead of "necessary"
L32-35: that in the North Sea *E. huxleyi* type A was dominant is stated twice
L46: "were of valley values" higher or lower?

Conclusions:
P19, L16: "…assemblages in cyclonic eddies were slightly productive." Productive in terms of what? This statement was not discussed.

Figures:

Figure 4. Please insert "percentage" after 40.

It was good to combine Figures 5 and 6 into Figure 6.

Can the new Fig. 7 show the different contributions of these species between Cyclonic and Anti-Cyclonic eddies?

---

## Referee Report (RR2)

The new version of the manuscript by Jin et al. substantially improved the discussion, which is clearer now. Some few comments are done from which only the one referring to page 10 is a statement that needs to be clarified.

**Specific comments:**

P8, L9: belonged to

P9, L1: Please consider changing « more » by « higher » or « more important »

P9, L25: Please consider changing « Some conceptions or terms » by « Some parameters »

P9, L27: Please consider changing « direct controlling factor» by « directly associated»

P10, L19-20: If P limitation usually increases coccolith size, why it is concluded That N imitation should be the main controlling factor on coccolith size?

---

## Editor Decision (ED1)

Referee #1

Lines 24-25 of Page 3; I think this is the first study that found E. huxleyi Type B from tropical-subtropical water, although several previous studies reported Type C or B/C from warm waters. I think it is worth to add some information on surface/vertical distribution of Type B E. huxleyi in the studied area.

Line 39 of the Page 3; coccolithophore not coccolith

Line 29 of the page 6; Spell out genus name of Gephyrocapsa ericsonii and Oolithotus fragilis.

Line 31-32; Reticulofenestra sessilis was reported from lower/deep photic zone of tropical-subtropical waters (Young et al. 2003).

Line 36 of page 6, Lines 5 and 22 of page 7; 'Western Pacific Warm Pool' or 'Tropical Warm Pool' not 'West Pacific Warm Pool'.

Line 21 of page 7 "morphotype A could be related to high sea-surface temperature (>26˚C) of the tropical SCS" ; How about the ecological preference/distribution of the Type B E. huxleyi observed in this study?

Lines 21-22 of page 7; Please add some references that explain definition/ distribution of the Western Pacific Warm Pool.

Line 2 of page 8; remove ')' after cyclonic.

Line 9 of Page 9, (Jin et al., in prep); I do not know if this journal allow to refer the manuscript in prep.

**Referee #2**
**General comments**

In most cases, replies to comments satisfactorily treated the general issues raised. However the discussion requires revision in the form. In some cases the changes in the manuscript discussion made it even more confusing (some examples are: P16 L42-45; P18 L14-19). Because of this lack of clarity in the discussion I believe the manuscript is not ready to be published.

According to the MS title, species composition and calcite content are the key parameters under study, but the largest part of the discussion goes to *Emiliania huxleyi* biometry (already in the introduction is E. *huxleyi* biometry and not calcite content the main parameters). The discussion of this section (4.4) goes back and forth around the same ideas and should be rewritten so it gets more concise.

Regarding the figures, I think the changes improved the MS.

In general, I believe the data set, calculations and statistical work do deserve publication but the discussion has to be improved. Readability should still be improved.

English is not my mother language so I did not focus on it, but a revision of grammar will be useful.

**Specific comments**

Abstract:
L15. Perhaps some words on how eddies regulated the coccolithophore community?

Introduction: I agree with the changes.

Methods:
P11, L22 and 23: Not necessary here.
L28: "In addition to" instead of "Apart from" or just starting the sentence in "Coccolith length"
P12, L34: Similarity matrix was constructed with biomass data? It was not explained how biomass was estimated.

Results:
P13, L35: "these features may indicate lateral transport" instead of "these features may be characteristics of either lateral transport". Perhaps the whole sentence should go to the discussion.
P14, L9: Italics for species name

Discussion:
P15, L15: I do agree it was probably related to the higher temperatures in the sampled area; but insist that this is not proved with the data of this study.
L38: Sentence needs to be re-written, I suggest: Results from nMDS, HCA and eddy settings in the 18°N section clearly showed that…..
P16, L29-31: thus, community composition did not change with environmental factors? In the previous section the differences between the results in cyclonic eddies, anti-cyclonic eddies, and adjacent stations were suggested to be driven by nutrients.

Same lines: It would be nice to elaborate more on the fact that *E. huxleyi* and *F. profunda* contributed more to water column calcite in cyclonic eddies than in anti-ciclonic eddies. This is not mentioned in the results section.

L42-45: I don´t understand "in anti-cyclonic eddies,… coccolithophore maximum layers were in group 3" does that mean that some stations in these eddies had a "coccolithophore maximum layer" while other stations did not? And that the first clustered together? Perhaps it refers to abundance instead of layer?

L45: "This transition" it should be clearer stated in the text

P16, L7: "coccolithophore community locations are similar with those in anti-ciclonic eddies" I don´t understand; does it means: environmental conditions or coccolithophores community composition/structure in those stations (I6 and I7) resembled those of AC eddies?.

L16-18: The last sentence of the paragraph is maybe not necessary as it was mentioned just before that POC fluxes are higher in the border of AC eddies.

L45: "coccolith calcite species" I understand what is meant but the term is confusing. It may be better stated as in the comment of the review and the response to this comment (P6, L26-28 in the Reply to referee #2)

P17, L1: please delete "totally"

P17, L9: I believe the whole section 4.4 should be re-written. The changes done so far did not help the MS. In this section, the argument that is proposed to explain the results stresses the importance of light limitation because nutrient limitation does not explain their results. However: 1) It is very difficult to follow the ideas that cut, come back, are exposed using confusing terms. 2) It goes back and forth around the same ideas: trying to explain why under P-limiting conditions cellular PIC generally has increased in culture experiments, although this is not "the core" of their arguments and could just be developed in one short paragraph. 3) PIC was not determined in this study but rather coccolith-coccosphere sizes and, although these parameters are obviously related, the author passes from PIC to coccolith size and to coccosphere size as equal without warning the reader.

L12: "coccoliths size has been show to change under low phosphate. .." how? Also, Engel et al. (2005) has been cited here and again in L2 of page 18 with the same purpose, can these paragraphs be synthesized into one?

L15: E. *huxleyi* calcite quota did not differ between the 6 strains tested in that experiment. The calcification rate did. Meaning growth rate changed in a strain specific fashion.

L16-17: This is not a novelty and it was not "revealed" by Aloisi´s model.

L17: "Phosphorus deficiency" instead of "phosphorus deficient". However, the whole sentence should be reconsider because Aloisi (2015) himself refered to Müller et al. (2008) (and others, please see page 4676). The same issue is raised again in page 18, the newly inserted L14-19. None of the two paragraphs (in page 17 or in page 18) are clearly written.

P18, L5: Müller et al. (2008), and Aloisi (2015) did not say that Engel´s et al. results meant that P limitation regulated growth rate…. This is the authors' interpretation.

L14 -19: It would be better to focous on PIC only and not to mention POC, since the whole discussion should go around coccolith biometry. Thus, parentheses are not necessary.

L19: "different perspective from coccolith calcification (rate)"?

L30: "possible" instead of "necessary"

L32-35: that in the North Sea *E. huxleyi* type A was dominant is stated twice

L46: "were of valley values" higher or lower?

Conclusions:

P19, L16: "…assemblages in cyclonic eddies were slightly productive." Productive in terms of what? This statement was not discussed.

Figures:

Figure 4. Please insert "percentage" after 40.

It was good to combine Figures 5 and 6 into Figure 6.

Can the new Fig. 7 show the different contributions of these species between Cyclonic and Anti-Cyclonic eddies?

---

## Author Response (AR2)

**Reply to Referee#1**

We would like to thank the anonymous reviewer for the pivotal comments to make the MS more rigorous.

Lines 24-25 of Page 3; I think this is the first study that found *E. huxleyi* Type B from tropical-subtropical water, although several previous studies reported Type C or B/C from warm waters. I think it is worth to add some information on surface/vertical distribution of Type B E. huxleyi in the studied area.

Reply: Really apologize for mistake. I have attributed all the non-type A morphotypes to "type B" as its derivative, as they are showing the similar features: delicate elements and non-grilled central area and the similar ecology: cold water preference. They mainly comprise type B, B/C and C, differentiated by their distal shield length. The morphotype we found can belong to type C according to Young et al. (2003), or type O? which is characterized by the open central area (Hagino et al., 2011). This mistaken has been revised in relevant sections.

[Figure]

Owing to the low magnification (×5000) of the SEM images got in NOC, I have re-taken some SEM images of the rare morphotype at higher magnification (×15000) to show the coccolith structure more clearly. The sample was obtained during the same cruise in the South China Sea. (B8, 50 m, 120 E 20.5 N)

Left: type C (O?), coccosphere

Right: type C (O?), detached coccolith

Line 39 of the Page 3; coccolithophore not coccolith

Reply: Revised.

Line 29 of the page 6; Spell out genus name of *Gephyrocapsa ericsonii* and *Oolithotus fragilis*.

Reply: Revised.

Line 31-32; *Reticulofenestra sessilis* was reported from lower/deep photic zone of tropical-subtropical waters (Young et al. 2003).

Reply: Agreement with Young et al. (2003), *Reticulofenestra sessilis* may dwell in relative deep layers of water-column. In the present case in the South China Sea, this species was sporadically found in some samples at the depth of 75 m.

[Figure]

SEM: *Reticulofenestra sessilis* and *Algirosphaera robusta* (H3 75 m)

LM: *Reticulofenestra sessilis* (H2 75 m, located at 19˚N 117˚E, another station in the same cruise, not included in the present study)

Line 36 of page 6, Lines 5 and 22 of page 7; 'Western Pacific Warm Pool' or 'Tropical Warm Pool' not 'West Pacific Warm Pool'.

Reply: Revised.

Line 21 of page 7 "morphotype A could be related to high sea-surface temperature (>26˚C) of the tropical SCS"; How about the ecological preference/distribution of the Type B *E. huxleyi* observed in this study?

Reply: Apologize for the mistake. This morphotype can be type C or O. Unfortunately, we cannot tell exactly the surface distribution of this morphotype, because it was sporadically found in some samples for its extremely low abundance. Actually, it was found in some samples at ~50 m, where was usually the coccolithophore abundance maximum depth in water-column. One possible reason could be that the high *E. huxleyi* cell density make it possible to find this rare morphotype. Alternatively, this morphotype has a vertical (temperature, nutrient) preference in water-column. As reported by Hagino et al. (2000), *E. huxleyi* type C was just found below the thermocline in the equatorial Pacific Ocean.

Lines 21-22 of page 7; please add some references that explain definition/distribution of the Western Pacific Warm Pool.

Reply: The Western Pacific Warm Pool is the ocean water mass, located in the western Pacific Ocean and eastern Indian Ocean, of which the sea-water temperature is consistently higher than 28 ˚C in a year round (Yan et al., 1992).

Line 2 of page 8; remove ')' after cyclonic.

Reply: Revised.

**References**

Young, J.R., Geisen, M., Cros, L., Kleijne, A., Sprengel, C., Probert, I., Østergaard, J.: A guide to extant coccolithophore taxonomy. International Nannoplankton Association. 2003.

Yan, X.H., Ho, C.R., Zheng, Q., Klemas, V. 1992. Temperature and size variabilities of the Western Pacific Warm Pool. Science, 258(5088), 1643-1645.

Hagino, K., Okada, H., Matsuoka, H.: Spatial dynamics of coccolithophore assemblages in the Equatorial Western-Central Pacific Ocean. Marine Micropaleontology, 39(1): 53-72. 2000.

Hagino, K., Bendif, E.L., Young, J.R., Kogame, K., Probert, I., Takano, Y., Horiguchi, T., des Vargas, C., Okada, H.: New evidence for morphological and genetic variation in the cosmopolitan coccolithophore *Emiliania huxleyi* (prymnesiophyceae) from the *COX1B-ATP4* genes. Journal of Phycology, 47, 1164-1176. 2011.

**Reply to Referee#2**

**General comments**

In most cases, replies to comments satisfactorily treated the general issues raised. However the discussion requires revision in the form. In some cases the changes in the manuscript discussion made it even more confusing (some examples are: P16 L42- 45; P18 L14 - 19). Because of this lack of clarity in the discussion I believe the manuscript is not ready to be published.

According to the MS title, species composition and calcite content are the key parameters under study, but the largest part of the discussion goes to Emiliania huxleyi biometry (already in the introduction is E. huxleyi biometry and not calcite content the main parameters). The discussion of this section (4.4) goes back and forth around the same ideas and should be rewritten so it gets more concise.

Regarding the figures, I think the changes improved the MS. In general, I believe the data set, calculations and statistical work do deserve publication but the discussion has to be improved. Readability should still be improved. English is not my mother language so I did not focus on it, but a revision of grammar will be useful.

We are grateful to the anonymous reviewer for the crucial comments to improve the quality of the MS. The section 4.4 concerning about the influences of nutrient and light on *E. huxleyi* coccolith size has been rewritten. The specific responses are posted as followed.

**Specific comments**

Abstract:

L15. Perhaps some words on how eddies regulated the coccolithophore community?

Reply: Yes, revised.

Introduction: I agree with the changes.

Reply: Thanks.

Methods:

P11, L22 and 23: Not necessary here.

Reply: Deleted.

L28: "In addition to" instead of "Apart from" or just starting the sentence in "Coccolith length"

Reply: Corrected.

P12, L34: Similarity matrix was constructed with biomass data? It was not explained how biomass was estimated.

Reply: The "biomass data" is the root-square-transformed coccolithophore absolute abundance data ($\sqrt{}$ (cell/ml)).

Results:

P 13, L35: "these features may indicate lateral transport" instead of "these features may be characteristics of either lateral transport". Perhaps the whole sentence should go to the discussion.

Reply: Corrected. It is also discussed in the section 4.3.

P14, L9: Italics for species name.

Reply: Corrected.

Discussion:

P15, L 15: I do agree it was probably related to the higher temperatures in the sampled area; but insist that this is not proved with the data of this study.

Reply: Agree. That different temperature preference of different morphotype of *E. huxleyi* is not supported in the present data, since the sampling was bounded by the tropical South China Sea in summer. Based on the previous works, we still made some short discussion concerning about the predominance of type A with its relation to high SST in the study area. The detailed examples have been deleted, and it is summarized as "In general … (see text P7. L24.) "

L38: Sentence needs to be re- written, I suggest: Results from nMDS, HCA and eddy settings in the 18 °N section clearly showed that…..
Reply: Corrected.

P16, L29 - 31: thus, community composition did not change with environmental factors? In the previous section the differences between the results in cyclonic eddies, anti-cyclonic eddies, and adjacent stations were suggested to be driven by nutrients.
Reply: The community composition did be controlled by environmental factors (light, nutrient). Here, we meant the bulk calcite concentration was not directly controlled by the environmental parameters of seawater, i.e. no statistical correlation. To avoid ambiguity, the sentence has been rephrased.

Same lines: It would be nice to elaborate more on the fact that *E. huxleyi* and *F. profunda* contributed more to water column calcite in cyclonic eddies than in anti-cyclonic eddies. This is not mentioned in the results section.
Reply: The Figure.7 was redrawn, based on which we rewrote these sentences to show the calcite contribution of different taxa in different oceanographic conditions with distinct coccolithophore community.

P15, L 42- 45: I don´t understand "in anti-cyclonic eddies …coccolithophore maximum layers were in group 3" does that mean that some stations in these eddies had a "coccolithophore maximum layer" while other stations did not? And that the first clustered together? Perhaps it refers to abundance instead of layer?
Reply: The "coccolithophore maximum layer" means the maximum coccolithophore abundance depth in water-column. In all stations, there was a coccolithophore abundance maximum depth which was correspondence with the DCM in water-column. Generally, the "maximum layer" was around ~50 m and belonged to the Group 2 (i.e. *E. huxleyi* dominated), however, in anti-cyclonic eddies, the "maximum layer" depth deepened (also correspondence with the DCM), the coccolithophore community in this layer belonged to Group 3. That is, *F. profunda* may represent the coccolithophore production in this subsurface layer. The sentence was rephrased.

P15, L45: "This transition" it should be clearer stated in the text
Reply: Rephrased, since the snap-shot like sampling we cannot tell exactly the transition process. However, it did be found that the different community composition were related to distinct oceanographic settings (i.e. eddy).

P16, L7: "coccolithophore community locations are similar with those in anti-cyclonic eddies" I don´t understand; does it means: environmental conditions or coccolithophores community composition/structure in those stations (I6 and I7) resembled those of AC eddies?
Reply: Sorry for ambiguity. Yes, this is what we mean. The sentence was rephrased.

P16, L45: "coccolith calcite species" I understand what is meant but the term is confusing. It may be better stated as in the comment of the review and the response to this comment (P6, L26 - 28 in the Reply to referee #2)
Reply: Thanks, revised.

P17, L1: please delete "totally"
Reply: Deleted.

P17, L9: I believe the whole section 4.4 should be re-written. The changes done so far did not help the MS. In this section, the argument that is proposed to explain the results stresses the importance of light limitation because nutrient limitation does not explain their results. However: 1) It is very difficult to follow the ideas that cut, come back, are exposed using confusing terms. 2) It goes back and forth around the same ideas: trying to explain why under P - limiting conditions cellular PIC generally has increased in culture experiments, although this is not "the core" of their arguments and could just be developed in one short paragraph. 3) PIC was not determined in this study but rather coccolith- coccosphere sizes and, although these parameters are obviously related, the author passes from PIC to coccolith size and to coccosphere size as equal without warning the reader.

Reply: The section 4.4 has been rewritten.

First, we discuss the *E. huxleyi* coccolith biometry. We think *E. huxleyi* size can well reflect its cell size and cell calcite content. The Fig.9 is added for discussion. Then we have made a caution that coccolith size cannot be compared with other calcification parameters. Thus, following discussion is confined to the coccolith/coccosphere cell size or PIC per cell variation in different environmental/experimental conditions.

Second, in the section 4.4.1 we discuss the influence of nutrient and light on coccolith size and try to interpret our data. In this section, we have removed the improperly comparison between *E. huxleyi* size and calcification rate in the previous MS, and many repeated statements (as the following comments have referred) and some unnecessary parts (as referred in the first vision of comments) were deleted.

L12: "coccoliths size has been show to change under low phosphate..." how? Also, Engel et al. (2005) has been cited here and again in L2 of page 18 with the same purpose, can these paragraphs be synthesized into one?

L15: E. huxleyi calcite quota did not differ between the 6 strains tested in that experiment. The calcification rate did. Meaning growth rate changed in a strain specific fashion.

L16 - 17: This is not a novelty and it was not "revealed" by Aloisi´s model.

L17: "Phosphorus deficiency" instead of "phosphorus deficient". However, the whole sentence should be reconsider because Aloisi (2015) himself referred to Müller et al. (2008) (and others, please see page 4676). The same issue is raised again in page 18, the newly inserted L14 - 19. None of the two paragraphs (in page 17 or in page 18) are clearly written.

P18, L5: Müller et al. (2008), and Aloisi (2015) did not say that Engel´s et al. results meant that P limitation regulated growth rate. This is the authors' interpretation.

14-19: It would be better to focus on PIC only and not to mention POC, since the whole discussion should go around coccolith biometry. Thus, parentheses are not necessary.

L19: "different perspective from coccolith calcification (rate)"?

Reply: Since, this whole section was rewritten. Please see details in the revised MS (red words in this section).

L30: "possible" instead of "necessary"

Reply: Corrected.

L32 - 35: that in the North Sea E. huxleyi type A was dominant is stated twice

Reply: Repeated content was removed.

L46: "were of valley values" higher or lower?

Reply: Revised. $\Omega_C$ and pH were low during Heinrich event 11.

Conclusions:

P19, L16: "…assemblages in cyclonic eddies were slightly productive." Productive in terms of what? This statement was not discussed.

Reply: As described in the results 3.2, community in Group 2 is characterized for the highest coccolithophore abundance and the dominance of r-select *E. huxleyi*, which could indicate coccolithophore is slightly productive in this group.

Figure 4. Please insert "percentage" after 40.
Reply: Corrected.

It was good to combine Figures 5 and 6 into Figure 6.
Can the new Fig. 7 show the different contributions of these species between Cyclonic and Anti-cyclonic eddies?
Reply: As the redrawn Figure 7 has shown the contributions of different species between different conditions, we may not have combined Figures 5 and 6, to show their results separately.